# Observed and modeled moulin heads in the Pâkitsoq region of Greenland suggest subglacial channel network effects

Celia Trunz[1,2], Kristin Poinar[3], Lauren C. Andrews[4], Matthew D. Covington[1], Jessica Mejia[3,5], Jason Gulley[5], and Victoria Siegel[6]

[1]Geosciences Department, University of Arkansas, Fayetteville AR, USA
[2]Center for Hydrogeology, University of Neuchâtel, Neuchâtel, CH
[3]Department of Geology and RENEW Institute, University at Buffalo, Buffalo NY, USA
[4]Global Modeling and Assimilation Office, NASA Goddard Space Flight Center, Greenbelt MD, USA
[5]Geosciences Department, University of South Florida, Tampa FL, USA
[6]Sisu Field Solutions LLC, Austin, Texas, USA

**Correspondence:** Celia Trunz (celia.trunz@gmail.com)

**Abstract.**

In the ablation zone of land-terminating areas of the Greenland Ice Sheet, water pressures at the bed control seasonal and daily ice motion variability. During the melt season, large amounts of surface meltwater access the bed through moulins, which sustain an efficient channelized subglacial system. Water pressure within these subglacial channels can be inferred by mea-
suring the hydraulic head within moulins. However, moulin head data are rare, and subglacial hydrology models that simulate water pressure fluctuations require water storage in moulins or subglacial channels. Neither the volume nor the location of such water storage is currently well constrained. Here, we use the Moulin Shape (MouSh) model, which quantifies time-evolving englacial storage, coupled with a subglacial channel model to simulate head measurements from a small moulin in Pâkitosq, western Greenland. We force the model with surface meltwater input calculated using field-acquired weather data. Our first-
order simulations of moulin hydraulic head either over-predict the diurnal range of oscillation of the moulin head or require an unrealistically large moulin size to reproduce observed head oscillation ranges. We find that to accurately match field observations of moulin head, additional subglacial water must be added to the system. This subglacial baseflow is likely sourced from basal melt and non-local surface water inputs upstream. We hypothesize that the additional baseflow represents strong subglacial network connectivity throughout the channelized system and is consistent with our small moulin likely connecting
to a higher-order subglacial channel.

## 1   Introduction

The Greenland Ice Sheet is experiencing increased mass loss via surface melting and calving in response to climatic warming (Hanna et al., 2020). In the ablation zone, most of the seasonal surface melt is routed through supraglacial streams (Yang and Smith, 2016; Pitcher and Smith, 2019) that drain into moulins (Smith et al., 2015), vertical shafts that penetrate the full
thickness of the ice sheet. Once meltwater reaches the ice sheet bed, it can change subglacial water pressure, which modulates ice motion. In this way, spatial and temporal variability in meltwater delivery to moulins can modulate sliding speeds on daily

to seasonal timescales. Spatial (Banwell et al., 2016) and temporal (Iken and Bindschadler, 1986; Bartholomaus et al., 2008; Schoof, 2010) variability in supraglacial meltwater delivery to moulins can accelerate or decelerate ice flow. The amount, position, and timing of meltwater infiltration into moulins determine local and regional ice motion, which in turn affects global sea-level change (Nienow et al., 2017).

The subglacial drainage system is composed of interspersed efficient and inefficient components (Iken et al., 1996; Mair et al., 2002; Röthlisberger, 1972; Walder, 1986). The efficient system is composed of low-pressure, high-flux subglacial channels, whereas the inefficient system is composed of a network of poorly connected, high-pressure cavities that conduct water at much slower speeds. While earlier studies suggested that the efficiency of the subglacial drainage system controls the seasonal pattern of velocities (Iken and Bindschadler, 1986; Bartholomew et al., 2010), field observations of moulin water level in Greenland have demonstrated instead the prominent role of the weakly connected drainage system (Andrews et al., 2014; Hoffman et al., 2016). As the melt season proceeds, the efficient channelized system grows in scope and can increase the connectivity of high-pressure cavities within the inefficient system, conducting water more efficiently overall and slowing ice motion (Andrews et al., 2014; Bartholomaus et al., 2011; Downs et al., 2018; Hoffman et al., 2016).

Diurnal ice motion cycles during the melt season are mainly influenced by the capacity of the channel-based efficient drainage to accommodate fluctuations in meltwater inputs (Bartholomew et al., 2012; Willis, 1995). Meltwater inputs to moulins initiate and sustain the growth of these subglacial channels, while the ice pressure drives creep closure when meltwater inputs subside (Schoof, 2010). Moulins directly feed these channels (Catania and Neumann, 2010; Yang and Smith, 2016); consequently, moulin hydraulic head (the water column height above a datum) reflects the water pressure in the local subglacial channel (Andrews et al., 2014). Despite their utility, field measurements of moulin hydraulic head fluctuations are sparse in Greenland (Andrews et al., 2014; Covington et al., 2020; Cowton et al., 2013; Mejia et al., 2021) and alpine glaciers (Iken, 1972; Badino and Piccini, 2002; Holmlund and Hooke, 1983; Vieli et al., 2004). This is due to the constraints of field logistics and the technical finesse required to instrument the complex, imperfectly vertical geometry that characterizes moulins (Covington et al., 2020; Gulley et al., 2009). Models for moulin heads are therefore needed if we are to understand diurnal water pressure variations across larger areas of the Greenland Ice Sheet in order to predict how meltwater infiltration affects ice motion (Trunz et al., 2022).

The most advanced subglacial hydrology models currently in wide use are two-dimensional models that simulate water pressure at any grid point as a continuum or a binary choice between two possible subglacial conditions: channels or cavities (e.g., Schoof et al., 2012; Sommers et al., 2018; de Fleurian et al., 2016). This type of model focuses primarily on pressures across the bed. Other two-dimensional models simulate only the channelized drainage system (e.g., Banwell et al., 2013). These two-dimensional models generally require a large number of parameters that often are unknown or uncertain. Alternatively, simpler physically based models have frequently been used to simulate water pressure in subglacial channels (Röthlisberger, 1972), with a subglacial channel that can melt open and creep closed (Spring and Hutter, 1981; Schoof, 2010), and is connected to cavities (Schoof, 2010; Bartholomew et al., 2012) or is not connected to cavities (Covington et al., 2012; Bartholomew et al., 2012; Cowton et al., 2016; Meierbachtol et al., 2013).

Some zero and one-dimensional models couple the subglacial channel to englacial storage via a cylindrical or conical moulin whose shape is static in time (Werder et al., 2013; Covington et al., 2012, 2020; Cowton et al., 2016; Bartholomew et al., 2012; Trunz et al., 2022), but not all models include such storage (de Fleurian et al., 2016). The size and shape of a moulin within the range of its water level oscillations affect the amplitude and temporal pattern of diurnal water pressure oscillations (Werder et al., 2010; Trunz et al., 2022). When these subglacial models are driven by realistic surface meltwater inputs, large englacial or subglacial storage volumes are required in order to produce realistic moulin head outputs (Hoffman et al., 2016; Bartholomew et al., 2012; Covington et al., 2020). To date, the true nature and location of these storage volumes have remained vague or the object of speculation (Flowers, 2018). For these reasons, we are motivated to investigate the size, shape, and water storage capacity of moulins, how these quantities change over time, and how these parameters affect subglacial water pressure and, therefore, moulin head variability.

In this study, we investigate the hydrodynamics in the englacial-subglacial system of a small single moulin in a moulin-dense catchment using a single-conduit subglacial model coupled with the Moulin Shape (MouSh) englacial hydrology model. This model allows the moulin to evolve in size, shape, and storage capacity in response to continued meltwater inputs. We compare modeled hydraulic head fluctuations with field measurements to infer characteristics of the englacial and subglacial systems.

## 2 Field site

As part of the Moulin Velocity Experiment (MoVE) project (Covington et al., 2020; Mejia et al., 2021), we collected moulin hydraulic head and supraglacial stream discharge measurements just upstream of an instrumented moulin during the 2017 melt season. The moulin was located at 69.4741°N, 49.8232°W, near "Low Camp" (Fig. 1a–b) in the Pâkitsoq region of the Greenland Ice Sheet. The site is approximately 25 km from the ice sheet margin at 780 m.a.s.l. where the ice is approximately 500 m thick (Morlighem et al., 2017). For simplicity, throughout the text we refer to these field features as the "moulin" and its "stream".

We collected moulin water levels every 15 minutes between July to October 2017 (Mejia et al., 2020b), however, we only use the moulin hydraulic head for a 40-day period during the melt season from mid-July through August 2017, after which surface meltwater input ceased. To ensure that the model and the measurements have the same point of reference, we use the bed at the moulin as the datum (we place the bed at 0 m), and we convert water pressure measurements to hydraulic head using the BedMachine-derived bed elevation and ice thickness at the moulin site.

## 3 Model and methods

We simulate the size, shape, and hydraulic head of a moulin instrumented in the field using the Moulin Shape (MouSh) model (Andrews et al., 2022) coupled to a subglacial channel model (Covington et al., 2020) based on the Schoof (2010) equations for melt and creep closure.

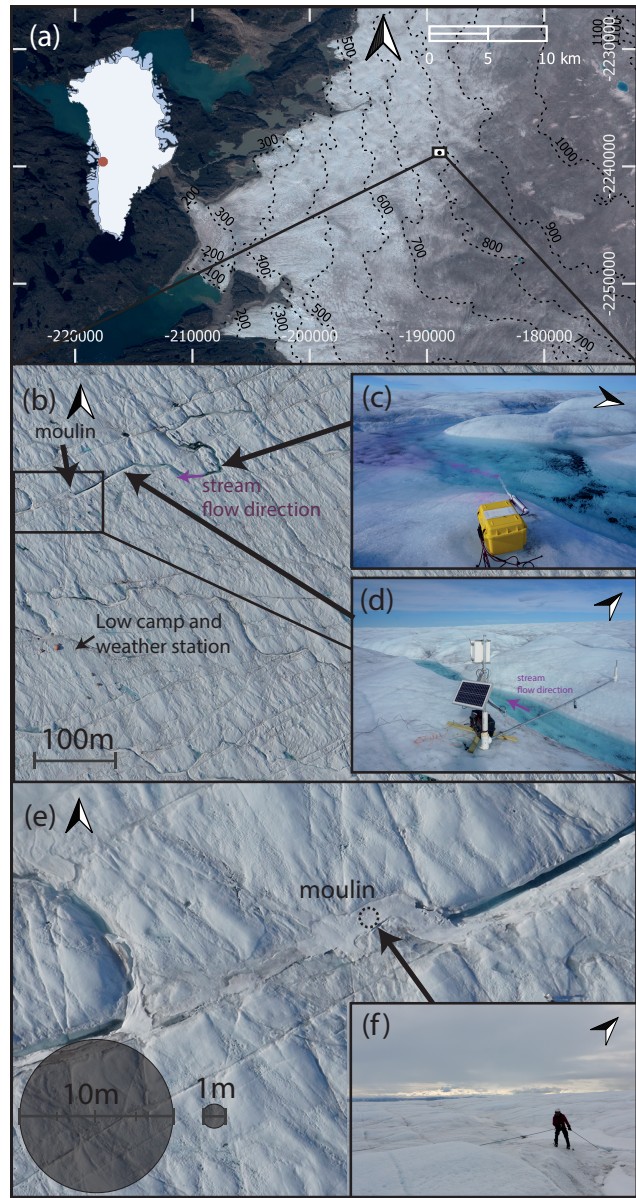

**Figure 1.** Study site. (a) Sentinel-2 satellite imagery from 2019 of the Pâqitsoq region in Greenland, with coordinates in the EPSG:3413-WGS 84/NSIDC Sea Ice Polar Stereographic North projection, in meters. The field site is indicated with the black box. (b) Orthophoto of the Low Camp field site produced with composite aerial imagery taken with a Tuffwing Uncrewed Aerial Vehicle (UAV) in July 2017. (c) Dye injection site. (d) Stream water level and dye measurement site. (e) Details surrounding the moulin. The arrow indicates the position of the cable where it disappeared under the one-meter wide snow cover on the stream and entered the moulin. (f) Pressure sensor lower in the moulin entrance.

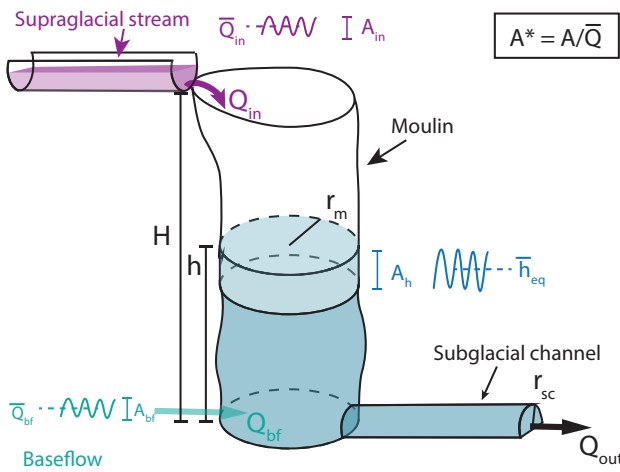

**Figure 2.** Sketch of the elements of the model, showing water fluxes ($Q_{in}$, $Q_{bf}$, $Q_{out}$), moulin and subglacial channel radii ($r_m$, $r_{sc}$), ice and water heights ($H$, $h$), and fluctuation amplitudes ($A_{in}$, $A_{bf}$, $A_h$).

All the components of the coupled moulin–subglacial channel model are illustrated in Fig. 2. We force the model with varying surface input with an amplitude ($A_{in}$) and mean discharge ($\overline{Q}_{in}$), which induces head ($h$) oscillations around the equilibrium head ($h_{eq}$) with an amplitude of oscillation ($A_h$). The water storage in the moulin is controlled by the co-evolution of the moulin radius ($r_m$) and the subglacial channel radius ($r_{sc}$). We also input a subglacial baseflow term ($Q_{bf}$) which prescribes the amount of water flowing in the subglacial channel without directly specifying flow or melt within the moulin. This baseflow can either be held constant or allowed to oscillate around a mean value ($\overline{Q}_{bf}$) with a peak-to-peak amplitude ($A_{bf}$), as qualitatively illustrated in Fig. 2.

### 3.1 Meltwater input

We force the model with a modeled and an idealized surface meltwater input ($Q_{in}$), based on the two days of discharge measurements (Fig. 3d) that we collected upstream of the moulin (Trunz et al., 2021), in order to extend the surface meltwater input time series to cover the entire melt season. First, we use a modeled surface input calibrated with the discharge measured in the field. Next, we generalize our stream observations using an idealized sinusoidal surface meltwater input. This sinusoidal forcing has an amplitude of oscillation $A_{in}$ and a mean $\overline{Q}_{in}$, as shown in Fig. 3.

### 3.1.1 Measured stream discharge

We measured the discharge ($Q_{in,meas}$) of the supraglacial stream approximately 100 m upstream of the moulin (Trunz et al., 2021), with a fluorescent dye dilution technique. We injected a Rhodamine WT solution with a peristaltic pump at a rate of 2 $\pm$ 0.5 mLmin$^{-1}$ ($Q_{pump}$). We measured the dye concentration (D) with a Turner Cyclops-7 submersible fluorometer calibrated

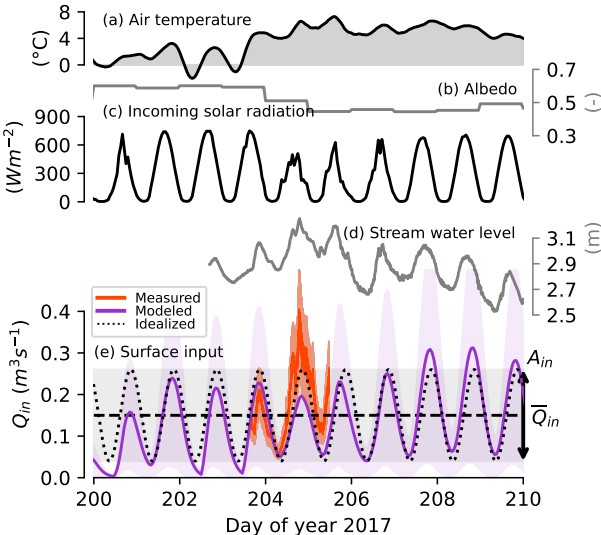

**Figure 3.** (a–d) Measured weather data used in the melt model from July 1–11, 2017. (e) Surface stream discharge as measured (Sect. 3.1.1), modeled (Sect. 3.1.2), and idealized (Sect. 3.1.3) for input into the MouSh model. Light purple shading indicates range of uncertainty in the meltwater input model without in situ measurements to calibrate it.

in the field and positioned 100 m downstream of the injection site. We used an injection concentration D of 200 ppb and calculated the stream discharge using

$$Q_{\text{in,meas}} = D \times Q_{\text{pump}}/S \tag{1}$$

Figure 3 shows the stream flow time series obtained over approximately two days. We used this record to calibrate the modeled discharge into the moulin.

### 3.1.2 Modeled stream discharge

To extend the discharge record beyond the short period of measurement, we modeled the surface melt ($M$) using an enhanced 110 temperature-index melt model (Pellicciotti et al., 2005) forced with meteorological measurements from a weather station co-located with the moulin (Mejia et al., 2020a).

We convert the melt rates ($M$, m/s) to runoff ($R$, $\text{m}^3\text{s}^{-1}$) as follows:

$$R = C_{\text{R}} M A, \tag{2}$$

where $A$ is the area of the internally drained basin, $0.24 \ \text{km}^2$ (Mejia, 2021), and $C_{\text{R}}$ is a runoff coefficient that we empirically 115 adjust to 0.9 to match the measured diurnal range of the stream discharge (Fig. 3).

To more accurately represent surface meltwater input, we add a routing delay to the runoff time series using a unit hydrograph transfer function, which has previously been utilized for a similar supraglacial stream in Greenland (King, 2018). We calculate

**Table 1.** Melt model parameters chosen from this study, compared with values selected by King (2018); Smith et al. (2017)

| Parameter | Units | This Study | King 2018 | Smith et al. 2017 |
|---|---|---|---|---|
| Drainage basin ($A$) | $m^2$ | 0.24 | 0.49 | 69-51.4 |
| Time to peak ($t_p$) | h | 2.5 | | 5.5–6.5 |
| Runoff coeff. ($C_R$) | – | 0.9 | 0.65 | 0.53–0.78 |
| Empirical coeff. ($C_p$) | – | 0.6 | 0.49 | 0.49–0.72 |
| Empirical exp. ($m$) | – | 1 | 2.1 | 2.1–4.6 |

the modeled meltwater input ($Q_{in,model}$) by convolving the modeled runoff ($R$) with a unit hydrograph *UH*:

$$Q_{in,model} = R * UH, \tag{3}$$

$$UH = \left(\frac{t}{t_p}\right)^m \left[e^{m\left(1-\frac{t}{t_p}\right)}\right] Q_p \tag{4}$$

We use the measured 2.5 h time to peak ($t_p$) calculated by Mejia et al. (2022) and empirically set the exponent $m = 1$, which is in the range of values used by King (2018) and Smith et al. (2017) and most accurately reproduces the minimum discharge values at our field area (Fig. 3). We calculate the peak discharge as $Q_p = C_p/t_p$, where $C_p$ is an empirical coefficient and for which we use the average value of 0.6 from Smith et al. (2017).

We calibrated the parameters $C_R$, $C_p$ and $m$ for both the runoff and the routing at the same time by visually comparing modeled discharge output with the measured stream discharge time series. The $C_R$ is slightly higher than at the Smith et al. (2017) sites elsewhere in western Greenland, where a range of values from 0.53 to 0.78 was found (Table 1), and compensate for potential underestimation of the drainage basin size $A$ due to the flatness of our field area.

Figure 3 shows the weather data used in the melt model (Fig. 3a–c), the observed stream water level (Fig. 3d), and the three surface meltwater input time series (Fig. 3e). The modeled meltwater input discharge agrees with the measured meltwater input with a root mean square deviation ($R^2$) of 0.7. The meltwater input model employs only three variables (air temperature, incoming shortwave radiation and albedo); therefore, it does not match all variations in the measured input. The largest discrepancy between measured and modeled stream discharge is apparent on day 205, which was cloudy; the model is known to underestimate melt during periods of cloud cover (Pellicciotti et al., 2005).

Melt models calculate an expected melt rate expressed in water equivalent height ($m/s$). Converting this to runoff ($m^3/s$) requires values for drainage basin size, ice density, and the fraction of water refreezing, which are minimally constrained parameters. Because the melt model underestimates melt during periods of cloud cover, we tuned $C_R$, $C_p$ and $m$ against measurements from day 204, a cloudless day. Parameter values that would best fit day 205 or 206 lead to a similar normalized diurnal surface input range (Sec. 3.1.4), which falls well within measured values across the Greenland Ice Sheet (Table 4).

The melt increase during day 205 is seen in the stream water level rise (Fig. 3d). The stream water level measurement is controlled by the cross-sectional area of the supraglacial stream, as well as the distance between the instrument and the bed of the stream, which can evolve through time. The latter effect is visible in our data (Fig. 3d) from day 206 and beyond: the stream deepened beneath our sensor , which was suspended from a pole and therefore fixed in elevation. The lowering trend of the stream water level after day 206 is therefore artificial.

Diurnal stream water level fluctuations (Fig. 3d) are in better agreement with the measured discharge ($R^2 = 0.94$) than with the modeled discharge ($R^2 = 0.23$), suggesting that the modeled surface input time series is imperfect in reproducing small variations of stream fluctuation throughout the day. However, it is the daily fluctuations that impact the moulin-subglacial channel the most and small variation smaller than a day or slight discrepancy between measured and modeled peaks do not matters in this case. Though the length of the measured time series we use to calibrate our stream discharge is only two days, the discharge measurements used to calibrate the model substantially reduce the range of the daily amplitude of oscillation in the model input $A_{\text{in}}$ relative to the mean discharge $\overline{Q}_{\text{in}}$ (Fig. 3e).

### 3.1.3 Idealized stream discharge

To separate the effect of diurnal fluctuations in surface input from weekly and seasonal variability, and to simulate a uniformly oscillating baseflow, we define an idealized sinusoidal surface input, $Q_{\text{ideal}}$:

$$Q_{\text{ideal}} = \frac{A}{2} \sin\left(\frac{2\pi(t + \phi)}{P_{\text{osc}}}\right) + \overline{Q}, \tag{5}$$

where $A$ is the peak-to-peak amplitude of oscillation, $P_{\text{osc}}$ is the period of oscillation (one day in this set of simulations), $t$ is the time, $\phi$ is the phase lag, and $\overline{Q}$ is the mean discharge.

For the idealized surface meltwater input $Q_{\text{in,ideal}}$ we assign $A_{\text{in,meas}} = 0.22 \text{ m}^3\text{s}^{-1}$ and $\overline{Q}_{\text{in,meas}} = 0.15 \text{ m}^3\text{s}^{-1}$ based on the field data shown in Fig. 3.

### 3.1.4 Normalized diurnal surface input range

To analyze the effect of supraglacial discharge variability on moulin hydraulic head variations, we introduce a parameter to quantify the relative amplitude of oscillations using a normalized diurnal input range, $A_{\text{in}}^*$:

$$A_{\text{in}}^* = A_{\text{in}} / \overline{Q}_{\text{in}}, \tag{6}$$

where $\overline{Q}_{\text{in}}$ is the mean supraglacial stream discharge and $A_{\text{in}}$ the peak-to-peak amplitude of oscillation.

### 3.1.5 Subglacial baseflow

The subglacial baseflow is a "black box" term added to the subglacial output $Q_{\text{out}}$; it does not transit through the moulin. Therefore, this baseflow influences only the moulin head and the subglacial cross-sectional area, and not the melting of the moulin walls. Like $Q_{\text{out}}$, baseflow also has units of volume per time. We use either a fixed or an oscillating baseflow in

our simulations. For the fixed baseflow, we vary the mean subglacial baseflow ($\overline{Q}_{\mathrm{bf}}$) from 0 to 5 $\mathrm{m^3s^{-1}}$. For the oscillating subglacial baseflow ($Q_{\mathrm{bf}}$), we also vary the amplitude of oscillation ($A_{\mathrm{bf}}$) from 0 to 0.44 $\mathrm{m^3s^{-1}}$ (Table 3).

## 3.2 Moulin Shape (MouSh) model

To simulate the shape of the moulin and the hydraulic head fluctuations, we use the Moulin Shape (MouSh) model (Andrews et al., 2022). The MouSh model is a two-dimensional physically based model that simulates the depth-dependent size and shape of a moulin for site-specific glacier properties (e.g., ice thickness, ice temperature, external stress, etc.) and for time-varying surface input forcing. MouSh initiates the moulin as a cylinder, then applies five components that enlarge or reduce the moulin radius at each point along a vertical axis, with the ability to melt the upstream and the downstream walls at different rates. These components are illustrated in Fig. 4. The deformation of the ice is simulated with a Maxwell viscoelastic model, where the instantaneous elastic deformation is independent from the time-dependent viscous deformation of the ice. In the model, the deformation components ("viscous" and "elastic") are primarily driven by the head, which counteracts the inward ice pressure. The melting of the wall occurs above ("open channel melt") and below ("underwater melt") the water level. Both components are strongly affected by the surface input rate. The horizontal creep of the ice under its weight is simulated with Glen's Flow Law, which uses ice thickness and surface slope. The surface slope (Table 1) is calculated along a flow profile going through the moulin area, which drops 400m over 50km (Mejia et al., 2021).

The MouSh model is coupled to a single subglacial channel model (Covington et al., 2020; Trunz et al., 2022) that implements the melting and creep closure equations for a subglacial channel from Schoof (2010). The cross-sectional area of the subglacial channel evolves depending on the hydraulic head and controls the retention and evacuation of the water in the moulin. The regional bed slope is assumed negligible due to the length of the subglacial channel and the scale of basal roughness.

With realistic combinations of ice thickness and surface input, MouSh predicts head positions consistent with the glacier geometry (i.e., head $h$ between the bed and the ice surface). For certain unusual combinations, MouSh predicts overflowing head, $h > H$, which is unrealistic and rarely observed in the middle of the melt season (an exception being by St Germain and Moorman, 2019, on a high Arctic mountain glacier). Therefore, we set up a threshold in the model that enforces $h \leq H$. Simulations can be run with or without this threshold.

Table 2 lists the values of all constants and tunable parameters we used in all MouSh simulations presented here.

## 3.3 Simulation categories

To investigate the controls on the observed moulin head oscillations, we test different representations of the englacial and basal hydrologic systems in our model. We compare modeled hydraulic head fluctuations and field observations (Sect. 2) to constrain the possible states of subsurface hydrology, using surface water inputs calculated from field data (Sect. 3.1.2) and constrained by stream discharge measurements (Sect. 3.1.1). In order to constrain how the subglacial drainage system and moulin interact to modify the amplitude of the moulin hydraulic head oscillations, we test different scenarios with fixed cylindrical moulin shapes (0.5 and 5 m radius) or evolving moulin shapes, in both cases with and without additional subglacial water input ("baseflow").

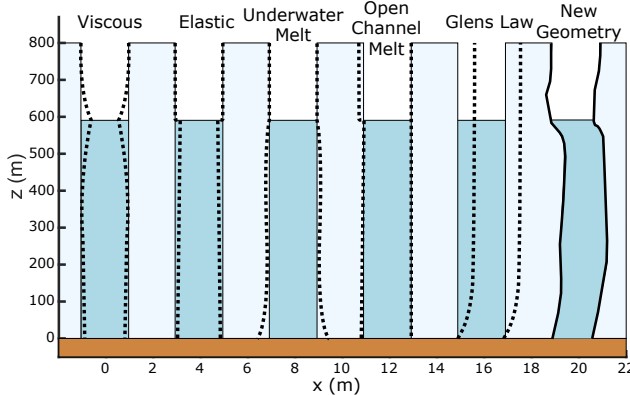

**Figure 4.** Vertical cross-sectional representation of the five components of the MouSh model: deformation (both elastic and viscous), melting (both turbulent underwater and open-channel subaerial), and lateral deformation by Glen's Flow Law. The horizontal position (x) varies along the moulin height above the bed (z). The first four components change the size and shape of the moulin. The final component changes only the shape. Sketch adapted from Andrews et al. (2022).

In Table 3 we list all the simulation names and parameters used in this study. We run simulations (Sims) driven by the field-observed stream discharge into a fixed cylindrical moulin (Sim F, for "Fixed") and a shape-evolving moulin (Sim E, for "Evolving"). For the shape-evolving moulin simulations, we run the model without baseflow (Sim EMa) and with baseflow (Sims EMb–e). For the simulations with baseflow, we assign a constant value or allow the baseflow to oscillate diurnally. We performed a wide range of tests and selected two constant baseflow values to present here: a large one of $2 \mathrm{~m}^3\mathrm{s}^{-1}$ (Sim EMb) and a small one of $0.5 \mathrm{~m}^3\mathrm{s}^{-1}$ (Sim EMc). For the smaller baseflow value $0.5 \mathrm{~m}^3\mathrm{s}^{-1}$, we add an oscillating component with a peak-to-peak amplitude of $0.2 \mathrm{~m}^3\mathrm{s}^{-1}$ that is either synchronous with the peak meltwater input (Sim EMd), or asynchronous, i.e. with a 12 h phase lag (Sim EMe). Simulations driven by modeled surface input begin on day 150, which provides a 50-day spin-up period.

Finally, we run three groups of simulations with idealized ("I") surface inputs to explore the modeled response to various (1) surface inputs (Sim EIa), (2) baseflow values (Sim EIb), and (3) surface input and baseflow phase lags (Sims EIc and EId). For each of these simulations with idealized parameter values, we calculate the diurnal range of moulin head variability after a period of ~23 days. This delay allows moulin head to equilibrate in the more extreme scenarios (Trunz et al., 2022).

**Table 2.** Constants and parameters common to all the simulations. The ice temperature profile FOXX 1 is from Lüthi et al. (2015).

| Constants | Value | Units |
|---|---|---|
| Ice density | 910 | $\mathrm{kg\,m^{-3}}$ |
| Water density | 1000 | $\mathrm{kg\,m^{-3}}$ |
| Gravitational acceleration | 9.8 | $\mathrm{m\,s^{-2}}$ |
| Latent heat of fusion | 332000 | $\mathrm{J\,kg^{-1}}$ |
| Water dynamic viscosity | $1.7916\times10^3$ | $\mathrm{Pa\,s}$ |
| Water thermal conductivity | 0.555 | $\mathrm{J\,(m\,K\,s)^{-1}}$ |
| Water heat capacity | 4210 | $\mathrm{J\,(K\,kg)^{-1}}$ |
| Ice heat capacity | 2115 | $\mathrm{J\,(K\,kg^{-1}}$ |
| Young's modulus | $5\times10^9$ | $\mathrm{GPa}$ |

| Parameters | Value | Units |
|---|---|---|
| Ice thickness, $H$ | 500 | m |
| Distance to margin, $L$ | $25\times10^3$ | m |
| Ice temperature, $T$ | FOXX 1 | °C |
| Regional surface slope, $\alpha$ | 0.01 | - |
| Initial moulin radius, $r_m(t=0)$ | 0.2 | m |
| Initial moulin head, $h(t=0)$ | 500 | m |
| Ice deformation enhancement factor | 3 | - |
| Basal ice softness | $6\times10^{-24}$ | $\mathrm{Pa^{-3}\,s^{-1}}$ |
| Moulin friction factor, $f_m$ | 1 | - |
| Subsurface friction factor, $f_{oc}$ | 0.5 | - |
| Subglacial friction factor, $f_{sc}$ | 0.1 | - |

## 4 Results

### 4.1 Simulations with realistic inputs

The results of our simulations are shown in Fig. 5. We simulate moulin head with surface inputs (Fig. 5a) calculated using the melt model (Sect. 3). We compare observations for simulations with fixed cylindrical moulins (Fig. 5b–c), evolving moulin shape (Fig. 5d–e), and evolving moulin shape with subglacial baseflow (Fig. 5f–g, and 6).

### 4.1.1 Fixed moulin shape

The results of the fixed cylindrical shape simulations (Sims Fa–c) are shown in Fig. 5b–d. The fixed cylindrical moulin with the smallest radius (Fig. 1e) of 0.5 m (Fig. 5b–c, lighter blue) produces head oscillations between three and four times the range

**Table 3.** Simulation names and parameters. Parameters are: initial subglacial channel radius ($r_{sc}(t=0)$), moulin radius ($r_m$), surface input type ($Q_{in}$), mean surface input ($\overline{Q}_{in}$), peak-to-peak amplitude of surface input ($A_{in}$), mean baseflow ($\overline{Q}_{bf}$), peak-to-peak amplitude of baseflow ($A_{bf}$), phase lag ($\phi$) between daily peak $Q_{in}$ and peak $Q_{bf}$. We name the simulation (Sim) according to its broad type: with a fixed (F) or an evolving (E) moulin shape, and whether the surface input is modeled (M) or idealized (I). To specify individual simulations within these broad types, we use lowercase letters (a, b, c, d, e).

| | $r_{sc}(t=0)$ | $r_m$ | $Q_{in}$ | $\overline{Q}_{in}$ | $A_{in}$ | $\overline{Q}_{bf}$ | $A_{bf}$ | $\phi$ |
|---|---|---|---|---|---|---|---|---|
| | shape | (m) | ($m^3s^{-1}$) | ($m^3s^{-1}$) | ($m^3s^{-1}$) | ($m^3s^{-1}$) | ($m^3s^{-1}$) | (h) |
| Sim Fa | 0.2 | 0.5 | Modeled | – | – | 0 | 0 | 0 |
| Sim Fb | 0.2 | 5 | Modeled | – | – | 0 | 0 | 0 |
| Sim Fc | 0.2 | 5 | Modeled | – | – | 1 | 0 | 0 |
| Sim EMa | 0.2 | Evolving | Modeled | – | – | 0 | 0 | 0 |
| Sim EMb | 0.6 | Evolving | Modeled | – | – | 2 | 0 | 0 |
| Sim EMc | 0.6 | Evolving | Modeled | – | – | 1 | 0 | 0 |
| Sim EMd | 0.6 | Evolving | Modeled | – | – | 1 | 0.2 | 0 |
| Sim EMe | 0.6 | Evolving | Modeled | – | – | 1 | 0.2 | 12 |
| Sim EIa | 0.6 | Evolving | Idealized | 0 to 1 | 0 to 1 | 0 | 0 | 0 |
| Sim EIb | 0.6 | Evolving | Idealized | 0.15 | 0.22 | 0 to 5 | 0 | 0 |
| Sim EIc | 0.6 | Evolving | Idealized | 0.15 | 0.22 | 0.2, 0.5, 1, 1.5, 2 | 0.22 | 0 |
| Sim EId | 0.6 | Evolving | Idealized | 0.15 | 0.22 | 1 to 2 | 0 to 0.44 | 0, 3, 6, 9, 12 |

of moulin head measured in the field. For our model to reproduce measured head oscillations (black lines), a fixed moulin radius of 5 m (Sim Fb) was required (Fig. 5b, dark blue). Even so, the moulin head oscillations are only within the range of measurements after day 220. Moreover, without baseflow, Sim Fb (the large 5 m moulin) produced a subglacial channel with a cross-sectional area of $\leq 0.3$ $m^2$, which is three orders of magnitude smaller than the $\sim 75$ $m^2$ size of the moulin directly above it. However, the addition of baseflow of $1m^3s^{-1}$ (Fig. 5b, dashed dark blue) yields a better reproduction of the head amplitude range measured in the field. In both cases without baseflow, the moulin head sits at the ice sheet surface through the beginning of the melt season, until day 214 (Fig. 5b).

## 4.1.2 Evolving moulin shape

Next, we show results of simulations where we allow the moulin shape to evolve via viscous deformation, elastic deformation, and wall melt (Sect. 3.2; Sim EMa). These model runs generate a moulin with a radius of ~0.5 m (Fig. 5e), which is comparable to surface observations of the moulin entrance, whose size is constrained by the 1 m width stream. The modeled moulin radius is also highly comparable to the subglacial channel radius of $\sim 1$ m (Fig. 6c). However, Sim EMa produces head oscillation amplitudes about six times larger than measured in the field. This large diurnal amplitude causes the simulated head to reach

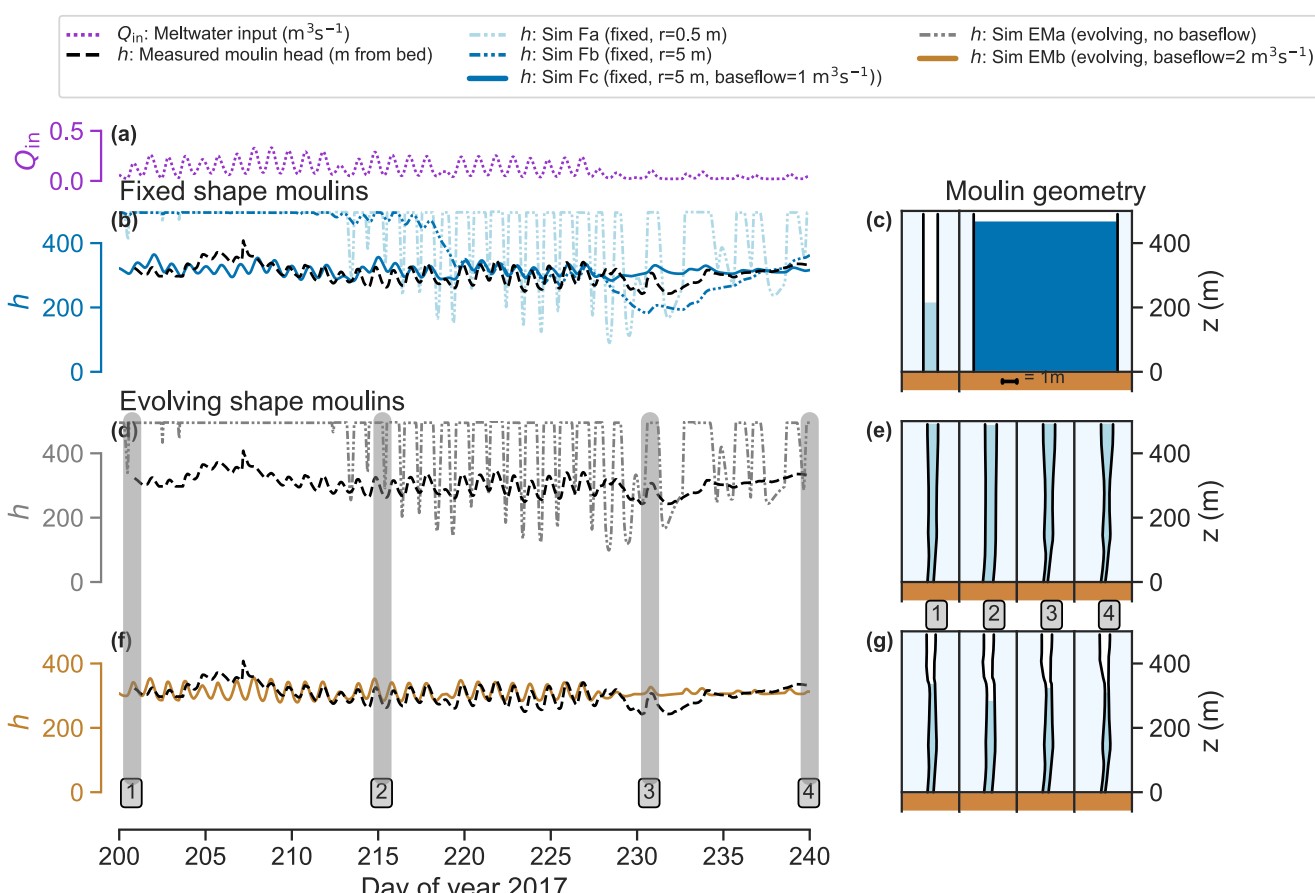

**Figure 5.** Comparison of three different simulations of moulin hydraulic head and moulin shape for the same surface input stream discharge ($Q_{in}$) calculated with melt model and scaled with discharge measurements represented in Panel (a) . Panels (b,c,d) show the measured and modeled moulin water levels (h). Panels (c,e,g) show the cross-sectional profile of the moulin as a distance above bed (z). Simulations Fa, Fb, and Fc: (b–c) Moulin shape is kept fixed with a radius of 0.5 m (left moulin and lighter tone) and 5 m (right moulin and darker tone), with and without baseflow. Simulation EMa: (d–e) Moulin shape is free to evolve through time. Simulated head plateaus are caused by head overflowing and model constraints that prevent water from rising higher than the ice thickness. Simulation EMb: (f–g) Moulin shape is free to evolve and an added baseflow reduces the head of oscillation. Four different timesteps are numbered and shaded in gray, corresponding to the moulin profiles in panels e and g.

the ice surface ($h = 500$ m) following high surface input events, similar to model runs from fixed-shape moulins with radii of 0.5 m. Head values this high are not supported by our field data.

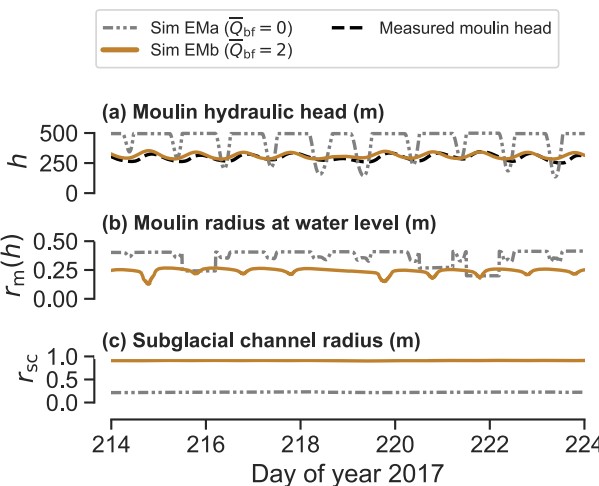

**Figure 6.** Comparison of (a) moulin hydraulic head, (b) moulin radius, and (c) subglacial radius for simulations with modeled surface input forcing, without subglacial baseflow (Sim EMa, dotted gray) and with $2 \mathrm{~m}^3\mathrm{s}^{-1}$ subglacial baseflow (Sim EMb, solid beige).

### 4.1.3 Effect of baseflow on simulated hydraulic head

We next run simulations with subglacial baseflow (Sims EMb–e). We find that a fixed $2 \mathrm{~m}^3\mathrm{s}^{-1}$ subglacial baseflow (Sim EMb,
Fig. 5f–g and Fig. 6b) significantly reduces the amplitude of oscillation of the hydraulic head without changing the order of magnitude of the moulin radius. The moulin radius is generally slightly ($\sim$10 cm) smaller for simulations with baseflow than without. This is because the head stays above overburden pressure for long periods of time in the no-baseflow simulation, decreasing the total amount of viscous and elastic closure.

While the general diurnal range of moulin head is reproduced with Sim EMb, the match between the simulated and measured
head is imperfect. We hypothesize three sources for this discrepancy: (1) limitations of the melt model in reproducing melt caused by particular weather conditions such as cloud coverage, which can underestimate melting, (2) subglacial lake drainages and rain events between days 205 and 210 (Mejia et al., 2021), and (3) the use of a constant value of baseflow, whereas subglacial flow conditions likely evolve throughout the season and enable the moulin water level to decrease at the end of the melt season (example during days 229 to 233).

For a given baseflow magnitude, an oscillating baseflow in phase with the surface meltwater input (Sim EMd) produces larger head oscillation amplitudes than when the baseflow is fixed (Sim EMc; Fig. 7). Additionally, we find that a 12 h phase lag (Sim EMe) in an oscillating baseflow of $0.5 \mathrm{~m}^3\mathrm{s}^{-1}$ produces a similar head oscillation amplitude as a constant, higher-magnitude ($2 \mathrm{~m}^3\mathrm{s}^{-1}$) baseflow (Sim EMb; Fig. 7c).

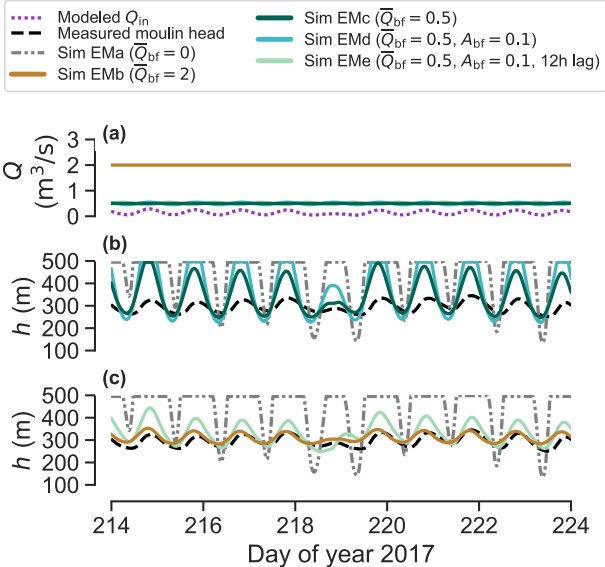

**Figure 7.** Influence of prescribed baseflow values on modeled moulin hydraulic head. (a) Modeled surface input used for simulations. (b) Moulin hydraulic head simulation for a mean baseflow of $1 \, \mathrm{m^3 s^{-1}}$, constant and with a peak-to-peak amplitude of oscillation of $0.22 \, \mathrm{m^3 s^{-1}}$. (c) Moulin hydraulic head amplitude with a constant $2 \, \mathrm{m^3 s^{-1}}$ and with oscillating $1 \, \mathrm{m^3 s^{-1}}$ with peak-to-peak amplitude of $0.22 \, \mathrm{m^3 s^{-1}}$ and a 12 h phase lag. Measured head is in black and simulated head without baseflow is in dotted gray.

## 4.2 Simulations with idealized surface inputs

Here, we investigate the relative effects of surface input (Sim EIa, Fig. 8a), baseflow magnitude (Sim EIb, Fig. 8b), and the phase lag between the baseflow and surface input (Sim EIc, Fig. 8b) on the hydraulic head oscillation dynamics. We use an idealized sinusoidal surface meltwater input and baseflow (Sect. 3) that enables us to control and compare the magnitude and amplitude of oscillations of both the inputs and the simulation outputs.

Figure 8a–b shows our model results without baseflow (Sim EMa) alongside the values of $A_h$ and $A_{in}^*$ measured for our
moulin (Mejia et al., 2021). We plot the mean as a red dot and bars for one standard deviation from the mean. This enables comparison between simulations made with idealized surface inputs and simulations made with the modeled surface inputs as well as field measurements. To calculate the mean and the standard deviation, we extracted $A_h$, the measured amplitude of head oscillation, and the modeled oscillation amplitude of the surface input from the simulations EMa and EMb.

In Sim EIb and EIc (Fig. 8b–c) we use a surface input representative of our observations, with a mean discharge of
$0.15 \, \mathrm{m^3 s^{-1}}$ and a peak-to-peak amplitude of $0.22 \, \mathrm{m^3 s^{-1}}$. The mean measured peak-to-peak head amplitude for the moulin during the middle of the melt season is approximately 10% of the ice thickness, and the mean simulated head with the modeled surface input, without baseflow, and without the ice surface threshold (Sect. 3) is in the range of 60% of the ice thickness.

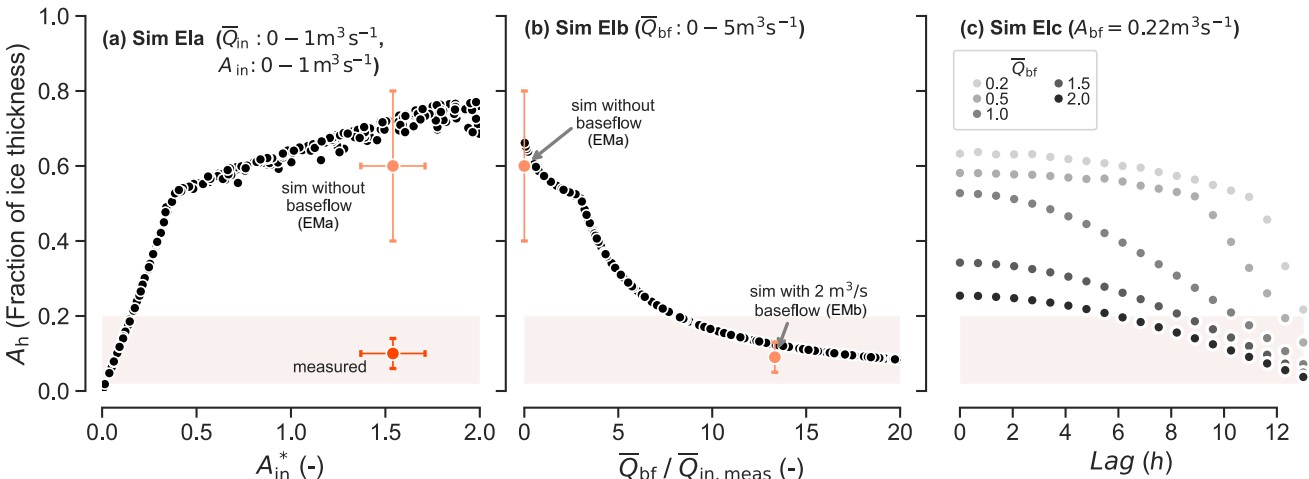

**Figure 8.** Diurnal range of moulin head ($A_h$) as a fraction of ice thickness, plotted against (a) Sim EIa, normalized diurnal range of the surface input ($A_{in}^*$), (b) Sim EIb, mean subglacial baseflow ($\overline{Q}_{bf}$) divided by the measured mean surface input ($\overline{Q}_{in,meas}$), and (c) Sim EIc, phase lag between surface input and baseflow. Because the head cannot flow above the ice thickness, head amplitude of oscillation $A_h$ larger than ~53% of the ice thickness are capped, producing a change of slope. (The location of this elbow is dependent on ice thickness; for H=500 m it is $A_h$ ~0.53.) The red dots are the mean values measured in the field and orange dots are the mean values from simulations with the field based modeled surface inputs, for comparison. Brackets represent variability of one standard deviation from the mean. Red shaded areas represent the field observed range of diurnal range of moulin head.

### 4.2.1 Effect of normalized diurnal range of surface input on simulated head

First, we investigate how the oscillation amplitude of moulin hydraulic head is affected by the normalized diurnal range of the surface input $A_{in}^*$ (Sim EIa, Fig. 8a). We simulate the moulin hydraulic head amplitude with selected values of mean surface input and peak-to-peak amplitude ranging from 0 to 1 $\mathrm{m^3 s^{-1}}$ (Table 3), and ice thickness of 500 m, the same as we estimate at our field site.

We find that when normalized by the mean discharge, the diurnal range of surface meltwater input strongly influences the simulated relative head amplitude ($A_h$). Specifically, $A_h$ steadily increase as $A_{in}^*$ increases from 0 to 0.3. For $A_{in}^* > 0.3$, the increase in $A_h$ slows. This is because when the diurnal head range is greater than $\sim 53\%$ of the ice thickness, moulin head reaches the ice surface during diurnal peaks. Because moulin water level cannot exceed the ice thickness, any further increases in the amplitude of head oscillations are produced by the lower minimum head values alone. Moulin head oscillates around the equilibrium head, and the distance between the equilibrium head and the ice surface determines the half amplitude of oscillation of the head before it gets capped. Thus, the specific position of the change of slope (elbow in Fig. 8a) depends on the ice thickness and the position of the equilibrium head.

Red and orange dots with bars in Fig. 8a show the relationship between $A_{in}^*$ and $A_h$ for Sim EMa (upper orange dot) and that measured in the field (lower red dot). Figure 8a demonstrates that, in the absence of baseflow, the amplitude of oscillation

relative to the mean (orange dot) must be ten times smaller than the measured value (red dot). The simulations with modeled surface input without baseflow (Sim EMa) predict a mean hydraulic head diurnal range of 60% of the ice thickness, while

field observations show oscillations over just 10% of the ice thickness. Indeed, none of the possible $A_{in}^*$ scenarios we tested produced $A_h$ results that are consistent with the field measurements.

### 4.2.2   Effect of constant baseflow on simulated head

Next, we investigate the effect of steady baseflow on the amplitude of diurnal moulin head oscillations (Sim EIb, Fig. 8b). We run simulations with idealized surface input consistent with our field observations (Fig. 3), with a peak-to-peak amplitude of

$0.22 \, \mathrm{m^3 s^{-1}}$ and a mean discharge of $0.15 \, \mathrm{m^3 s^{-1}}$. For those simulations, the baseflow values ranges from 0 to $5 \, \mathrm{m^3 s^{-1}}$. For our catchment of $0.24 \, \mathrm{km^2}$, $5 \, \mathrm{m^3 s^{-1}}$ is the equivalent of twenty additional moulins being fed the same surface melt upstream of our moulin, directly connected to a subglacial channel that connects exclusively with the subglacial channel under our moulin. This compares to moulin density lower than $1/ \, \mathrm{km^2}$ often observed in western Greenland (Banwell et al., 2016).

We find that, in order to reduce the head oscillations into a realistic range (Sim EIb, Fig. 8b), a constant baseflow of at least

eight times the mean surface input is required. When the baseflow is less than four times the mean surface input, we observe an unrealistically high head amplitude, and when the baseflow is higher than six times the mean surface input, the diurnal range of moulin head approaches observed values. To reproduce the observed diurnal range for Sim EMb (Fig. 8b, red dot), a constant baseflow of $2 \, \mathrm{m^3 s^{-1}}$, which is about fourteen times the mean surface input, is required. Simulation EIb (Fig. 8b) shows that the sensitivity of the diurnal range in head to baseflow magnitude greatly reduces beyond approximately eight times the mean

surface input.

### 4.2.3   Effect of surface input–baseflow phase lag on simulated head

Finally, we investigate the effects of an oscillating baseflow on the time evolution of moulin head. We experiment with different magnitudes and phases of oscillation relative to the surface input $Q_{in}$ (Fig. 8c and Fig. 9). Figure 8c shows the diurnal range in head with a surface input similar to that observed in the field with an oscillating subglacial baseflow with the same amplitude of

oscillation ($0.22 \, \mathrm{m^3 s^{-1}}$) as the surface input. We vary mean subglacial baseflow from 0.2 to $2 \, \mathrm{m^3 s^{-1}}$ with a phase lag ranging from 0 to 12-h. With a 12-h phase lag, baseflow values of at least triple the mean surface input are required to dampen moulin head oscillations to match the observed range. On the other hand, with a 6-h phase lag, baseflow fifteen times larger than the mean surface input is required to match observations.

Figure 9 illustrates the effects of baseflow on diurnal head range. Specifically, five different surface input–baseflow phase

lags from 0 to 12 h are shown. The resulting $A_h$ for each run is represented in percent $A_h(A_{bf} = 0)$ when simulated with no baseflow. As baseflow increases, the moulin head oscillation decreases at different rates for different lags, until the amplitude of oscillation of the baseflow, $A_{bf}$, equals the measured amplitude of oscillation of surface input, $A_{in,meas}$. When $A_{bf}$ becomes larger than $A_{in,meas}$, the head oscillation amplitude rises again, at a rate controlled by the amplitude of baseflow oscillations. Thus, we observe an increase in relative $A_h$ as $A_{bf}/A_{in,meas}$ increases above 1. We also find that when the baseflow and the

meltwater amplitude and magnitude are identical ($A_{bf}/A_{in,meas} = 1$) and antiphased (lag of 12 h), the head amplitude drops to

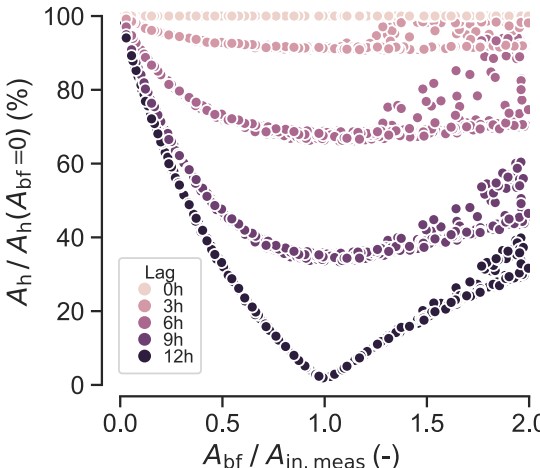

**Figure 9.** Sim EId: Hydraulic head oscillation amplitude $A_h$, as a percent of the amplitude without baseflow ($A_h(A_{bf} = 0)$), versus the ratio between baseflow amplitude ($A_{bf}$) relative to the measured surface meltwater input amplitude($A_{in,meas}$), for different surface input–baseflow phase lags. Baseflows with larger amplitudes of oscillations (x-axis), or that are more out of phase with the surface melt (darker colors), generally produce smaller-amplitude head oscillations.

zero. As another example, a lag of 6 h with $A_{bf}$ at least half of $A_{in,meas}$ reduces the moulin head amplitude to 70 % of its value with a zero-lag baseflow.

## 5   Discussion

In this study, we provide the first comparison of modeled hydraulic head in a shape-evolving moulin to direct field measure-
ments of a small subglacially connected moulin in Greenland. This enables us to scrutinize the relative roles of measured surface meltwater input and hypothesized subglacial water fluxes on a field-observable quantity, moulin head. From this comparison, we infer general traits of regional subglacial connectivity.

We identify three potential controls on the diurnal range of moulin hydraulic head: (1) the normalized diurnal range of the surface meltwater input ($A_{in}^*$) , (2) the shape of the moulin in the head oscillation range, and (3) the addition of a baseflow. The
first control acts as the primary driver of moulin head fluctuations, while the other two are filters that dampen or amplify the surface meltwater input signal (Covington et al., 2020; Trunz et al., 2022). We discuss each of these potential controls in the subsections below.

### 5.1   Effect of surface input on head oscillation range

Our simulation results in Fig. 8a show that a substantially smaller daily oscillation range of meltwater input ($A_{in}^* \lesssim 0.2$) than
observed in the field is required to produce moulin head oscillations in the measured range. While the snow cover earlier in

**Table 4.** Normalized diurnal surface input ($A_{in}^*$) and its constituent properties, mean input ($\overline{Q}_{in}$) and range of amplitude ($A_{in}$), observed at supraglacial streams on the Greenland Ice Sheet.

| Study | $\overline{Q}_{in}$ (m$^3$s$^{-1}$) | $A_{in}$ (m$^3$s$^{-1}$) | $A_{in}^*$ (-) |
|---|---|---|---|
| Chandler et al. (2013) | ~2 | ~3.5 – 4 | 1.8 – 2 |
| McGrath et al. (2011) | ~0.2 | ~0.3 – 0.4 | 1.5 – 2 |
| Marston (1983) | ~0.08 | ~0.1 – 0.16 | 1.3 – 2 |
| Muthyala et al. (2022) | ~0.3 | ~0.3 – 0.6 | 1 – 2 |
| Smith et al. (2017) | ~15 | ~15 – 20 | 1 – 1.3 |
| This study | ~0.15 | ~0.15 – 0.3 | 1 – 2 |

the melt season might reduce the diurnal variability of $A_{in}^*$, middle to late melt season measurements of supraglacial discharge elsewhere around Greenland show normalized diurnal range of the surface input $A_{in}^*$ from 1 to 2 (Table 4). This is similar to what we find at our site. The compilation of measurements in Table 4 shows that the daily peak-to-peak oscillation of surface input is usually larger than, and generally at least comparable to, the mean discharge in supraglacial streams entering moulins.
Equivalently, Table 4 shows that surface inputs generally halve ($A_{in}^* = 1$) and can even drop to near zero ($A_{in}^* \sim 2$) at their overnight minima.

While moulin size is the dominant control on head oscillation range when considering a wide range of fixed moulin sizes (Covington et al., 2020), we find a clear relationship between the normalized amplitude of input $A_{in}^*$ and the diurnal range of moulin head $A_h$ (Fig. 8a) when the moulin size is constrained by a model. Thus, when the moulin shape is known or estimated,
the mean discharge through a supraglacial stream ($\overline{Q}_{in}$), and its diurnal oscillation ($A_{in}^*$), are central to predicting daily pressure fluctuations.

## 5.2 Moulin storage as a source of damping

Moulin sizes and shapes in Greenland are poorly constrained by field evidence to date. In our simulations, the radius of the moulin is determined by the MouSh model (Sect. 3.2), which provides an estimate of the moulin size and shape in the portion
of the moulin where the water level fluctuates, and allows us to isolate the effect of surface input variability on moulin head oscillation amplitudes (Trunz et al., 2022).

Simulation of moulin head using the field-observed normalized diurnal range of surface input, $A_{in}^*$, produces extremely large diurnal ranges in head (Sim EMa, Fig. 5d), which overwhelm the model. Similar behavior has been observed in other modeling studies (Cowton et al., 2016; Bartholomew et al., 2012; Werder et al., 2010). Reduction in hydraulic head variability
can be obtained by increasing the moulin size, which is inherently uncertain due to limited field exploration to date. For the same surface input forcing, larger moulin volumes will produce more damped oscillations than narrower moulins (Trunz et al., 2022). While large moulin storage volumes may absorb strong variations in surface inputs (Covington et al., 2020; Trunz

et al., 2022), for the small-input moulin we study here, the cross-sectional areas would have to be at least ten times larger than predicted by the MouSh model and observed at the surface (Fig. 1e) for this to occur.

### 5.2.1 Potential biases of the MouSh model

Our simulated moulin cross-sectional area varies between $\sim 0.3$–$0.5$ m$^2$ at and below the water line. This is of the the same order of magnitude as the subglacial channel ($\sim 0.2$–$1$ m$^2$). Since moulin size is potentially an important source of damping of the moulin head variability (Trunz et al., 2022), we explore here the potential for the MouSh model to underestimate moulin volumes.

In the recent development of the MouSh model, (Andrews et al., 2022) results for typical Greenland inputs and ice parameters yielded moulins with radii $\sim 0.5$–$1$ m at the water line, approximately the size of the moulin we study here. Modeled moulins are larger than this above the water line by a factor of $\sim 2$–$5$ (Andrews et al., 2022).

The moulin geometry simulated by MouSh is particularly sensitive to the ice softness and friction factors of the moulin walls. Softer ice tends to decrease the moulin volume, as does high friction, which increases melt along the moulin walls (Andrews et al., 2022). MouSh uses a friction factor, a relatively unconstrained parameter, to govern this wall melt.

At and below the water line, an increase of the friction factor by one order of magnitude increases the moulin radius by only a factor of two. Unrealistically stiff ice would be required to influence the moulin radius by the same amount. To act as a low pass filter and damp the diurnal fluctuations of the head water level fluctuations, the moulin radius would have to be at least an order of magnitude larger than the radius simulated by the MouSh model.

Despite its sensitivity to the friction factor, melt rates below the water line are simpler, better constrained, and less variable than those above the water line Recent field exploration of Greenland moulins discovered cavernous chambers (Covington et al., 2020; Reynaud and Moreau, 1994) where the cross-sectional areas are larger beneath the surface than at the entrance and remain large up to depths of approximately $100$ m. The cross-sectional area of the Phobos Moulin (Covington et al., 2020), the largest moulin explored to date, is about ten times larger at a depth of $70$ m than it is at the surface. The explored cross-sectional areas deep within FOXX Moulin (Covington et al., 2020) and Isortoq Moulin (Reynaud and Moreau, 1994) are at most about twice the cross-sectional area at the entrances. Thus, large volumes similar to the one present in Phobos Moulin may not be typical of all Greenland moulins. When it was explored in 2018, Phobos had been active for at least two years; in each melt year, the stream feeding it flowed in from a slightly different direction, melting the walls from different sides, which, among other things, could have enlarged the near-surface chamber. Inventories of moulins in this same area suggest that reuse for 2–3 years is common among lake-draining moulins (Poinar and Andrews, 2021; Andrews, 2015), however, the stream typically flows in from a consistent direction. MouSh model runs over a single melt season (Andrews et al., 2022) are able to predict neither this size nor the overhung shape observed in Phobos Moulin.

Thus, there is a distinct possibility that in certain cases, our MouSh results underestimate the volume of the moulin above the water line by a factor of 5–10. The presence of a hypothetical larger upper chamber connected to the water level fluctuation zone in our moulin would reduce the need for baseflow by a factor of three (Sect. 4.1.1) if the volume of the moulin at the water level was ten times larger than the moulin radius at the surface (see comparison of moulin radius in Fig. 1e). However

because our instrumented moulin is a recently opened moulin fed by a relatively small stream (Smith et al., 2017), we consider it unlikely that such a large volume would be present 150 m below the surface.

### 5.2.2  Other sources of englacial or supraglacial storage

Subglacial hydrology models have dealt with extreme head fluctuations produced by the large normalized diurnal range of surface input observed in the field by using larger moulin volumes. On an alpine glacier, Werder et al. (2010) and Schuler and Fischer (2009) both required a moulin radius two to three times larger than they expected in order to damp the simulated head observations. Clarke (1996) used a lumped model with several reservoirs to do the same. Overflowing and overpressurization of the subglacial system is not unique to single-conduit subglacial models. Larger values of englacial void ratios ranging from

$1 \times 10^{-4}$ to $1 \times 10^{-2}$ (Downs et al., 2018; Hewitt, 2013; Werder et al., 2013) or temporary subsurface storage (Hoffman et al., 2016) are often required in dual distributed–channelized models to prevent overflowing or overpressurization of the subglacial system. Considering a moulin density at our site of 10 moulins per $\mathrm{km}^2$ (Mejia, 2021), and assuming a 1 m moulin radius, we obtain an englacial void ratio of $3 \times 10^{-5}$. This is 1–3 orders of magnitude smaller than the values used in the models mentioned above. Hence, moulin storage may not represent the only storage site that allows the damping of subglacial water pressure.

The MouSh model does not account for potential subsurface storage outside the moulin. Such storage has been used to dampen head oscillations in other models via various methods. For example, Bartholomew et al. (2011) use a circular reservoir with a radius 80 times larger than their simulated moulin to temporarily store water and prevent overflow. Cooper et al. (2017) hypothesized the weathering crust as a significant storage reservoir, but we are able to dismiss this possibility for our study area because our measurements of the surface input were taken immediately upstream of where the stream enters the moulin

(Fig. 1). As another example, Hoffman et al. (2016) simulated moulin head measured in a similar field area to ours and assumed that when the moulin head was above 60 m below the ice surface, the water was stored in crevasses or fractures that would slowly release it back into the moulin or provide for additional moulin storage in cases where multiple large crevasses intersect each others. In our case, moulin head never reaches this height; at its highest, our measurements show it $\sim$100 m below the surface (Fig. 5).

Deeper englacial storage close to the head oscillation range could influence the moulin water level dynamics. Based on our fixed cylindrical moulin simulations (Sim Fb), we find that to substantially influence the water level fluctuation, the cross-sectional area of the englacial void would have to be about 80 $\mathrm{m}^2$ (e.g. a crevasse 10 m long and 8 cm wide), reaching a depth of 150-200m. However, at the surface we only observe small centimeter-scale-width of the fractures. Comparing the width:depth aspect ratio of a fracture to the estimated material properties of ice (shear modulus $\sim$ 1 GPa) and surface deviatoric stresses

at our field site ($\leq 10 - 100$ kPa), we expect crevasse depth in our area to be around 10 m, but no more than 100 m (Poinar et al., 2017). While water storage in deep fractures cannot be completely ruled out, crevasses within this field area are unlikely to be sufficiently open to these depths. However, other form of englacial storage could exist.

In temperate glaciers, englacial conduits following fractures and cut-and-closure formation reported by (Gulley et al., 2009) could produce englacial storage are formed on longer timescales. However this process has not been observed in our study

area, which has cold ice ($\sim$ -10°C). Heterogeneous hydrofracture processes could also lead to englacial storage. For example,

a few kilometers away from our field site we observed multiple moulins along a line that could be connected with each other through a larger crevasse, and some did have visible crevasses associated with them. However, at our site, there is no surface crevasse passing through or close to the moulin (Fig. 1e). In addition, Andrews et al. (2022) show that any moulin formation process shorter than two weeks is unlikely to influence the moulin shape.

## 5.3 Basal processes as a source of damping

At our site, the damping of moulin head variability cannot be satisfactorily explained by supraglacial or englacial storage (Sect. 5.2.2). We therefore next speculate how basal processes could influence moulin head variability.

Sensitivity analysis of the subglacial channel model performed by Andrews et al. (2022) showed that increased basal softness of the ice would decrease the moulin head variability by enabling more reactive ice melt and creep. During low surface input periods, the subglacial channel would creep closed faster, and during high flow periods it would melt open faster. However, explaining the observed variations would require unrealistically soft basal ice: about six orders of magnitude softer than ice at $0°C$.

More likely, our results highlight the limitation of modeling moulin head in a closed system. Our model comprises just a single moulin that connects to a single subglacial channel. In reality, as shown in Figure10, multiple moulins within a local area feed a complex subglacial hydrologic network that includes multiple well-connected channels, more isolated areas, and distributed cavities. The model we use does not allow us to simulate those complex processes. However, it gives a first-order estimation of the importance of basal processes in the damping of moulin water level. Accordingly, our addition of baseflow to our model approximates the role of these other systems, especially other moulins and a dendritic channel network.

### 5.3.1 The requirement for baseflow

Two versions of our simulations produced results that matched field observations: Sims Fb and Fc, a $5$ m radius static moulin with and without baseflow respectively, and Sims EMb and EMe, size- and shape-evolving moulins with subglacial baseflow. In the preceding sections, we explored possible scenarios that could produce the englacial storage volumes that would effectively be equivalent to the large moulin in Sim Fb. We found that none of these scenarios were realistic. Here, we explore possible sources of subglacial water flow represented by the baseflow term introduced to our model (see Sims EMb and EMe).

The simulations without baseflow (e.g., Sim EMa) produce a subglacial channel that is too small ($0.05$ m$^2$) to have enough discharge capacity to evacuate the water when the head increases at the beginning of each melt day (Fig. 6). This has the result of overfilling the moulin, with the head unrealistically exceeding the ice thickness by midday every day (Fig. 5). Every afternoon, as the discharge in the stream decreases, the water in the moulin is rapidly evacuated through the subglacial channel, which has opened during the day and does not immediately constrict as the surface input reduces. Instead, this closure proceeds overnight under the resulting low moulin head, again producing a small subglacial channel that gets overwhelmed by increasing melt volumes the next morning. Dow et al. (2014) found similar modeling results and concluded that subglacial channels are unlikely to exist under thick ice.We find that increasing the water flux through the subglacial channel makes the water flux in the subglacial channel more constant, preventing the nightly creep closure of the channel. This has the effect of increasing the

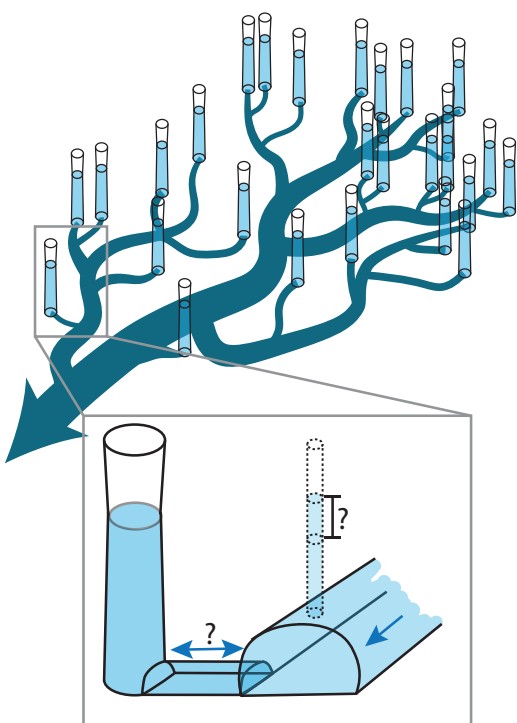

**Figure 10.** Subglacial network connectivity illustrated by a dendritic subglacial channel system (dark blue), initiated by moulins forming tributaries that connect to higher-order channels.

size and capacity of the subglacial channel, and making it more resilient to diurnal changes in surface input. This increase in water flux is incorporated to our modeled system by the introduction of baseflow to the subglacial channel. Importantly, the inclusion of this baseflow term was the only change or addition to the model that achieved agreement between modeled and observed moulin head.

### 5.3.2 Subglacial channel network connectivity and baseflow

Our simulations implement baseflow as a direct addition to the subglacial channel directly upstream of the moulin. This is a first-order modification to our simple, single-channel model that is necessary to make the moulin hydraulic head oscillation agree with field data (Fig. 7). While some moulins may be directly on the path of a subglacial channel and receive significant additional subglacial inputs directly upstream of moulins (Christoffersen et al., 2018; Hoffman et al., 2018; Werder et al., 2013), others initiate the subglacial channel when a crevasse intersects a supraglacial stream (Andrews, 2015; Benn et al., 2009; Holmlund, 1988). Water exiting moulins that do not feed an existing subglacial flowpath will flow towards a higher-order (larger) subglacial channel (Gulley et al., 2012). Because multiple moulins feed into this higher-order subglacial channel, the discharge within this channel will be larger than what enters our moulin. Thus, this higher-order channel may carry the baseflow

that prevents the moulin head from dropping when surface inputs wane, thereby keeping the moulin-connected subglacial channel large enough to readily evacuate the subsequent day's meltwater inputs.

Figure 10 shows a simplified conceptualization of the network connectivity providing the baseflow. Water pressure within the higher-order channel will be controlled by all the moulins feeding it upstream. If the water pressure in the moulin exceeds that in the higher-order channel, water will evacuate the moulin through the tributary. Conversely, if the water pressure in the moulin is lower than in the higher-order channel (for example, when the moulin head is at its nightly low, but the higher-order channel is still in the process of constricting), this will temporarily reverse the hydraulic gradient and provide water input back into the first-order tributary channel, preventing the moulin head from dropping. This flow reversal or external basal flow has been observed in boreholes (Holmlund and Hooke, 1983; Gordon et al., 2001) and moulins (Mejia et al., 2021). Overall, the higher-order channel has a stabilizing effect on the head in the tributary and, consequently, in the moulin.

This inference is supported by measurements of normalized diurnal water output ranges at glacier outlets by Bartholomew et al. (2012) and Cowton et al. (2016), which range from 0.1 to 0.5. This is considerably lower than the range in moulins ($A_h \sim 0.4 - 0.8$, Fig. 8), suggesting that diurnal pressure fluctuations in higher-order channels are damped by the network connectivity. Cowton et al. (2016) show that flow in a higher-order channel prevents the head from dropping when surface input decreases. Although they only allow flow in the direction from the moulin to the higher-order channel, their conceptual model would allow the water pressure there to exceed that in the moulin during daily low-input periods in the case of a sustained or lagged higher-order channel discharge. This would be consistent with our concept of temporary flow reversal from the higher-order channel back to the moulin.

Additionally, the subglacial channel model assumes that there is no bedslope. While this assumption is reasonable for the entire subglacial channel scale, it might not hold locally. For an isolated moulin, flatter areas tend to decrease local hydraulic gradients and reduce basal flow in subglacial channels, while steeper areas increase local hydraulic gradients and enable faster evacuation of subglacial water. Therefore, changes in local gradient caused by interconnected moulins cannot be reflected with our single subglacial channel model. Those local changes may drive part of the moulin water level dynamics.

### 5.3.3 Baseflow in the lower and upper ablation zones

Our results show that either a large, constant baseflow or a smaller, oscillating baseflow out of phase with surface input can dampen the diurnal range of moulin head to match our field observations. With both the surface input and moulin shape constrained in our modeling setup, we find that a constant subglacial baseflow of 2 $\mathrm{m^3 s^{-1}}$, nearly 15 times larger than the mean surface input (Sim EMb), or an antiphased 1 $\mathrm{m^3 s^{-1}}$ oscillating baseflow with half the amplitude of oscillation of the surface input (Sim EMe), brings the moulin hydraulic head amplitude of oscillation into the observed range. Which scenario is more likely will vary with location on the ice sheet, as we explain next.

At our study site, we anticipate the discharge in the higher-order subglacial channel connecting to the instrumented moulin to be relatively constant, given the large range of surface input phase lags and magnitudes upglacier from the moulin investigated here (Mejia et al., 2022). We observed at least ten other moulins in a 1 km radius around the moulin in 2017 and 2018 (Mejia, 2021) with relatively small drainage basins (<1 $\mathrm{km^2}$); these all likely feed a common higher-order channel, as we

illustrate in Fig. 10. Inputs from nearby moulins with similarly sized drainage basins will increase the discharge in the higher-order subglacial channel and generally stack to produce a daily oscillation in baseflow, similar to Sim EMd. Moulins further upstream, where drainage basins are larger due to thicker ice and lower surface gradients (Yang and Smith, 2016; Andrews et al., 2022), likely also feed this same channel. Input from these moulins, however, will dampen oscillations in the subglacial

baseflow by adding out-of-phase discharge to the channel. This would be more consistent with Sim EMb.

Higher on the ice sheet, where the moulin density is much smaller (Phillips et al., 2011; Poinar et al., 2015), we expect a lower water flux through the higher-order subglacial channel, potentially low enough to be more consistent with Sim EMd or Sim EMe (low-flux, oscillating baseflow) than Sim EMb (high-flux, non-oscillating baseflow). If the baseflow oscillation is in phase with the surface input, then it would tend to increase the amplitude of head oscillations (Fig. 7, Sim EMd), whereas

field observations suggest that this amplitude actually decreases in thicker ice (Covington, 2020). Thus, we hypothesize that high-elevation moulins are few enough in number that the lag in surface input from one moulin to another should produce the required baseflow oscillations (Sim EMe). This contrasts with low-elevation moulins, whose greater number density and variation in lag should produce a large, minimally oscillating baseflow in the main subglacial channels, similar to Sim EMb.

### 5.4 Potential external source of baseflow

Spatially discrete moulin inputs are necessary for initiating and sustaining efficient subglacial drainage (Dow et al., 2014). While the damping of the hydraulic head for a single moulin can be attributed to its subglacial network connectivity with other moulins, we consider here other potential external sources of water: sources aside from surface meltwater coming from the ablation zone in the drainage area.

A recent study shows that fast subglacial water flow velocities recorded with tracer tests require an additional non-local

source of subglacial flow (Chandler et al., 2021). This is consistent with our simulation results and with our foremost hypothesis that baseflow originates from the subglacial network connectivity, where moulins are connected to other moulin inputs (Fig. 10) through an arborescent system of subglacial channels (Davison et al., 2019).

Previous studies found suggestions of water storage in the englacial and subglacial system (Chu et al., 2016; Poinar et al., 2019; Rennermalm et al., 2013) that could provide a seasonal or year-round water source upstream of our moulin. This could

potentially sustain larger subglacial conduits, even for moulins at the upstream edge of the ablation area. In addition, during high water pressure events in moulins, water can be pushed out of the subglacial channel to the surroundings and back to the channel when the pressure in the moulin decreases (Andrews et al., 2014; Mair et al., 2003; Gordon et al., 1998; Hubbard et al., 1995). This could provide for an anti-phased baseflow.

Subglacial meltwater is also produced through basal melting. We estimate that our studied moulin connects to an upstream

subglacial catchment of $\sim$2000 $\mathrm{km}^2$, based on subglacial drainage basins calculated by Mankoff (2020), Mankoff et al. (2020), and the local basal melted–frozen boundary estimated by Poinar et al. (2015). This is a much larger area than our supraglacial catchment size (0.24 km$^2$ – four orders of magnitude difference) since it extends 400 km inland. The geothermal gradient representative of western Greenland is $\sim$50 $\mathrm{mW/m^2}$, which produces basal melt rates of $\sim$0.05 m/a (Fahnestock et al., 2001; Downs et al., 2018), which gives a total basal melt flux of $\sim$3 $\mathrm{m^3 s^{-1}}$. This is very similar to the subglacial baseflow we require

in our simulations (2 m$^3$s$^{-1}$, Sect. 4.1.3). More recent calculations of basal melt by Karlsson et al. (2021) account for all basal energy sources, not just the geothermal source we use here. Their results of ∼0.05 ma$^{-1}$ in our study area agree with our calculation above.

Our estimated catchment-integrated basal water flux of 3 m$^3$s$^{-1}$ is an upper bound because it assumes that all basal melt reaches the subglacial channel system connected to our moulin. In reality, before it can make it to the subglacial channel system, some portion of this available basal melt is stored in the inefficient portion of the drainage system with only a small portion of the water traveling through the linked cavity system (Andrews et al., 2014; Kingslake and Ng, 2013), and some is lost through the bed to the groundwater system (Vidstrand, 2017). At the same time, this will be compensated by surface meltwater inputs from moulins upstream (Sect. 5.3.2). Based on surface melt climatology, we expect the rates of upstream surface water inputs to the higher-order subglacial channels to be larger than the fluxes lost to the inefficient and groundwater systems (Vidstrand, 2017).

## 6    Conclusions

Our results suggest that the moulin we instrumented requires larger inputs than surface meltwater alone to keep its subglacial channels large enough to accommodate the observed wide diurnal range of surface input. The observed diurnal range of hydraulic head inside a moulin cannot be explained by local constraints (surface input, moulin size and local ice sheet properties) alone, but requires other, non-local water inputs to the subglacial system. With surface and shallow subsurface external inputs dismissed at our site, this additional water is most likely basal in nature. Local sources of basal water, such as basal melt or groundwater, are unlikely to be large enough to reduce the moulin head amplitude of oscillations. Instead, this requirement for additional subglacial input is best explained by a strong connectivity of the moulin and its subglacial channel to a network of subglacial channels fed by other moulins. This connectivity, or non-local control on moulin hydraulic head and local basal water pressure, suggests a complex subglacial hydraulic network, consistent with other work. Our results provide additional new evidence of the importance of the connectivity of moulins through a subglacial network. Finally, our results suggest that subglacial water flow in main channels is lower-magnitude and has stronger daily oscillations at higher altitudes under thicker ice, whereas at lower altitudes under thinner ice, it is likely steadier. This difference in subglacial water flux is likely to affect ice motion at higher altitude differently than closer to the margin in future climates, when surface meltwater inputs increase.

*Code and data availability.* Hydraulic head data for the moulin (called JEME in the data set) can be found on the Arctic Data Center website at doi:10.18739/A2M03XZ13. Meteorological data used for in the melt model can be found at doi:10.18739/A2CF9J745. Model simulations were produced with the python version of the MouSh model (pyMouSh). The current version of the Github repository containing the pyMouSh model is the Release v.1.0.0 and is available here: https://zenodo.org/record/7058365

*Author contributions.* CT, KP, and LCA conceived the ideas; CT performed model runs, created the figures, and implemented the unit hydrograph; CT wrote the manuscript with guidance from KP; JM and CT did the data curation; JM implemented the melt model and provided melt time series; JM, CT, MDC, VS, and JG designed and coordinated the field effort and collected the field data; MDC and JG acquired funding; All authors reviewed and edited the manuscript.

*Competing interests.* At least one of the co-authors is a member of the editorial board of *The Cryosphere*. The peer-review process was guided by an independent editor, and the authors also have no other competing interests to declare.

*Acknowledgements.* C.T., M.D.C., J.M., and J.G. were supported by the United States National Science Foundation grant 1603835. K.P. and L.C.A were supported by NASA Cryosphere grant 80NSSC19K0054. The GNSS base station and on-ice stations were provided by UNAVCO in collaboration with National Science Foundation Logistical support was provided by CH2MHill Polar Services. We thank Charlie Breithaupt, Ronald Knoll, Brandon Conlon and others for their assistance in the field.

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
