# Peer review of "Observed and modeled moulin heads in the Pâkitsoq region of Greenland suggest subglacial channel network effects"

_The Cryosphere, 2022_

## Author Comment (AC1)

**Author response to comment on "Observed and modeled moulin heads in the Pâkitsoq region of Greenland suggest subglacial channel network effects"**

Response to Samuel Doyle, Referee #1

**The authors' replies are in bold blue**

**General Comments**

Well-written, timely, and based on a substantial body of recent work this manuscript would make an excellent contribution to The Cryosphere and the topic in general. The manuscript presents the results from a moulin-channel model - the methods for which are described in a previous study (Andrews et al., 2022). The model does not include any glacier hydrological systems other than a single moulin and a single Rothlisberger channel and depending upon your perspective this represents the main strength or the main weakness of this study. The conclusion that there is additional "base flow" that contributes to and damps flow within moulin-connected channels is uncontroversial given that many moulins occur on the Greenland Ice Sheet and many appear to be connected (e.g. Andrews et al., 2014). However, this does not take away from the contribution this study makes in very neatly explaining observed moulin head variability using a numerical model. The assertions made regarding differences in "base flow" in the lower and upper ablation areas (Section 5.3.3) are intriguing and demonstrate recent advances in our understanding of the moulin-connected drainage system.

The main finding of this manuscript is that channel growth is too slow - and channels are therefore too small - to explain the observed damping of diurnal moulin head variations, when the model is forced by local moulin inputs alone. This finding is somewhat similar to that in Dow et al. (2014) but a different interpretation is given in that study: that channels are unlikely to form or persist at high elevations. Current evidence (e.g. Covington et al., 2020; Chandler et al., 2021) as well as the modelling presented in this manuscript suggests channels do form at high elevations and that connectivity to other channels may help them persist. Can the discussion of this be expanded slightly?

> **We thank Samuel Doyle for the thorough review and for all the useful comments. We are pleased to see that the summary above describes the findings in the manuscript well. We will expand the discussion on the subglacial channel at higher elevations with the Chandler et al. (2021) reference suggested, which is indeed independent corroborating evidence to what we find (the markings of non-local inputs to the channelized subglacial network).**

**Specific Comments**

L36 - Water pressure in moulins was also measured by Holmlund & Hooke (1983) and Vieli et al. (2004).

    **We added the two references.**

L396 - Section 5.3.2 and Sections 5.3 and 5.4 in general - As mentioned above, a limitation of this study is its application of a moulin/channel model without including other components of the subglacial drainage system. This is stated clearly in Section 5.3 (L375) but isn't mentioned in Section 5.3.2 which deals only with subglacial channels. It would be clearer if Section 5.3.2 was renamed from "Subglacial network connectivity and base flow" to something more specific to channels e.g. "Subglacial channel network connectivity and base flow".

    **We renamed the Section as suggested.**

Note that filling of a moulin from the base by subglacial water was observed by Holmund & Hooke (1983) and has been observed in boreholes by several studies (e.g. Gordon et al., 2001) as well as being the focus of Mejia et al. (2021). This suggests that "reverse flow" into moulins does occur and is unlikely to be limited to flow within channels. I appreciate that the model cannot reproduce this - and I don't suggest this is attempted in this paper - but can this limitation be discussed alongside the direct evidence listed above?

    **We discuss the reversal of flow caused by a change of hydraulic gradient in Section 5.3.2. We added "This flow reversal or external basal flow has been observed in boreholes (Gordon et al., 2001; Holmlund & Hooke, 1983) and moulins (Mejia et al., 2021) ."**

Section 5.3 appears to overlap with Section 5.4. Is Section 5.3 only concerned with damping caused by storage within the unchannelised system and not damping caused by recharge from the unchannelised system? Can this be made clearer? Can these sections be combined?

    **We changed the title of Section 5.3 as suggested above. Section 5.3 does focus on the channelized system. As each of these sections are quite lengthy already (Section 5.3 is three pages with 3 subsections, and Section 5.4 is a full page), we prefer to keep them separate for better signposting and smaller topics.**

In the Section 5.4 heading what does "external" relate to? Is it the same as "non-local"? Does it mean from an unchannelised system.

    **External does relate to non-local. However this does not specifically mean unchannelized system. "External" describes sources aside from surface meltwater in the ablation zone - e.g. basal melt, or multi-annually stored water (e.g. in the unchannelized basal system, or in the englacial system). "Non-local" refers to faraway (>10 km?) meltwater from the ablation zone. We will clarify this in the manuscript by formally defining it as we have done here.**

As in previous studies (Dow et al., 2014; Meierbachtol et al., 2014) Shallow surface and bed slope have a critical role in channel development. How was the slope ratio of 0.01 (equivalent to ~0.6 degrees) measured? Is the bedslope assumed to be the same as the surface slope in Table 1? If so, state this in the methods. If you were to increase the surface and/or bedslope would this change the results?

**The surface slope is calculated along a flow profile going through the moulin area, which drops 400 meters over 50 km (slope of 0.01) (see Figure 1 in Mejia et al. 2021). The slope does locally vary, but this value is a good representation of the whole area. The surface slope in the model influences the moulin shape, but not the hydraulic gradient. It is true that in reality, the local hydraulic gradient will be influenced by the surface topography. However, on a larger scale, we assume that it is mainly controlled by the length of the channel and the ice thickness. In the subglacial channel model, the bedslope from the moulin to the margin is negligible considering the distance to the margin compared to the actual bed variation.**

**More locally, in the moulin area, it is true that the bed slope is more variable. For isolated moulins, flatter areas will likely decrease local hydraulic gradients and reduce basal flow in subglacial channels, while steeper areas will enable faster evacuation. We will add this point in the Discussion Section 5.3.2.**

L128 - Stating the p-value for correlation as a measure of accuracy is inappropriate. The low p-value suggests that the correlation between modelled discharge and measured water level is unlikely to be due to random variation. The p-value does not tell us the degree of accuracy as a statistically significant correlation is plausible for any variables that co-vary regardless of the magnitude (or units) of the variables, or whether there is a causal relationship. For the same reason the statement of "agreement" on L124 between the same variables as above is not strictly speaking supported by the coefficient of determination given, which is a measure of correlation rather than accuracy, though this depends on the intended meaning of "agreement" which is relatively vague. Strictly speaking, the coefficient of determination of 23% suggests 23% of the modelled discharge can be explained by variation in stream level. Reporting the correlation between measured stream level and discharge may be useful but it should not be described as accuracy. A better measure of accuracy would be the root mean square deviation between modelled and measured discharge. To be clear, I see no problem with the modelled discharge imperfectly matching the measured discharge given how difficult it would be to model rainfall and turbulent heat fluxes.

**This is a good point, we will calculate the root mean square deviation between modeled and measured discharge. We will also modify section 3.1.2 to correct our statistical analysis and interpretation, and expand on the relative accuracy of the modeled meltwater input in section 3.1.2.**

L129 - Can you state that m and c are linear regression coefficients and state which variables are the subject of the regression? Is this the melt model calibration mentioned on L123? It's unclear why day of year 205 was included in the regression even though it was affected by

rainfall. Overall, this section needs expanding and revising, possibly with the addition of a figure showing the linear regression.

**m and C_R are not results from linear regression, they are empirically set values (Equations (2) and (4)) that we tuned manually. We reorganized section 3.1.2 to make it more clear, improved the naming convention of certain parameters, and made a new table (Table 1) containing the parameters used for the modeled stream discharge. We removed the mention to the rainfall event, which actually happened in 2018 but is not covered in this manuscript. Instead, the discrepancy between the measured melt and the modeled melt is caused by the cloud coverage. This is the reason why the melt model is calibrated manually. We will expand on this in the method section.**

L216 - can you revise this sentence? The melt model reproduces melt not "particular weather conditions". It's unclear what is being underestimated. If correct, can you revise to make it clearer that melt is underestimated by the model under cloudy conditions and consider adding an appropriate citation.

**We restated the sentence to "...limitations of the melt model in reproducing melt caused by particular weather conditions such as cloud coverage, which can underestimate melting … "**

L230/L251, Fig. 8 and Fig. 8's caption - consistent description of the error bars would help the reader.

**We fixed the description. This was not accurate. The bars represent 1 standard deviation from the mean. Not errors.**

L244 - Is "ranges" required here because you refer to the ranges in amplitude or is range already implicit in amplitude?

**Removed "ranges". Ah already refers to an amplitude.**

Fig. 9 is difficult to interpret. The y-axis label gives two versions of $A_h$ with the modifier in brackets meaning different things (0 relates to amplitude without baseflow, while n relates to time lag). Could $A_h(0)$ be written as $A_h$ minus $A_{bf}$? A different symbol for time lag to n, which is usually sample size, would be more intuitive. The symbol for time lag needs to be used consistently (e.g. in the text, legend and caption). A simpler y-axis label could be used and defined in the caption e.g. something like "normalised diurnal head amplitude". The plot could be labelled with arrows showing where on the x-axis $A_{bf}$ exceeds $A_{in}$ and vice versa.

**We will improve Fig. 9 and integrate the reviewers comment.**

L274 and other occurrences - consider using the past tense for things that were done in the past.

**We will do as suggested.**

L367 - This is the orthodox view that crevasses are unlikely to be sufficiently open below a few tens of metres below the surface but there is not that much observational evidence to support this view. It is worth remembering that the moulin would have originated from water flowing into a fracture and that englacial conduits often follow such fractures (e.g. Gully, 2009). I don't think storage within englacial fractures can be definitively ruled out. See also evidence for fractures at depth presented in Hubbard et al. (2020) and evidence for energy released by refreezing meltwater which potentially occurs in fractures (Luthi et al., 2015).

**We agree that water storage in deep fracture cannot be conclusively ruled out. However, to substantially influence the water level fluctuation, the cross-sectional area of the englacial void would have to be about 80m$^2$, so for example a crevasse 10 m long and 8 cm wide, reaching a depth of 150-200m. At our specific site, there is no visible crevasse passing through or around the moulin (see Figure 1e). Also, the general topography of the area suggests that we are more in a compressive zone, while the Hubbard et al. (2021) study is situated in a faster flowing region that is more likely to be in extension. In addition, Andrews et al. (2022) show that any moulin formation process shorter than two weeks is unlikely to influence the moulin shape. Englacial conduits following fractures have been found in temperate glaciers, and cut-and-closure formation reported by Gulley et al. (2009) are formed on longer timescales and could produce englacial storage, but this process has not been observed in our study area, which has cold ice (~ -10°C). The refreezing observed by Luthi et al. (2015) took place over multiple centuries (Poinar, 2016) as the ice advected through areas of various stress states- in contrast to the seasonal or sub-seasonal storage we seek here. We will discuss potential englacial planar storage in the moulin in Section 5.2.**

L451 - reference Figure 10

**Referenced as suggested.**

L452 - discrete in what sense, temporal or spatial or both?

**Changed to "Spatially discrete .. "**

L459 - Nienow et al. (2005) comes to this conclusion by interpreting velocity data on the basis of pressure measurements presented elsewhere (e.g. Hubbard et al., 1995; Gordon et al., 1998). It may be better to cite the studies with hydrological observations of this process.

**Modified as suggested.**

L484 - Can you reconcile the assertion that at lower altitudes subglacial water flow is steadier while at higher altitudes it is lower in magnitudes but greater in diurnal oscillation amplitude, with the opposing observations in Covington et al. (2020)? Perhaps this would apply if fixed moulin and channel sizes were assumed. Could you expand this assertion in the conclusions slightly to better reflect how it is described (very nicely) on ~L445?

**We changed "in the main subglacial channel" to "This contrasts with low-elevation moulins, whose greater number density and variation in lag should produce a large,**

**minimally oscillating baseflow in the main subglacial channels, similar to Sim~EMb.”
in Section 5.3.3 to avoid confusion. We expect the steadier flow in the main subglacial
channels fed by multiple moulins, while the observations made by Covington et al.
(2020) are from moulins and not subglacial channels. The lower diurnal range in head
compared to a lower altitude moulin could be caused for example by damped surface
meltwater inputs, and connectivity with a moulin with very different phase lag. We will
expand this assertion in the conclusion, as suggested.**

**Technical Comments**

L17 - Move citation beginning “(Yang …” to before “that”

    **Moved as suggested.**

L74 - delete second citation to Morlighem et al. (2017)

    **Deleted as suggested.**

L90 and other occurrences - check whether “Fig. 3” should be “Figure 3” when referenced in the
main text outside of brackets.

    **The rule is ‘The abbreviation "Fig." should be used when it appears in running text and
    should be followed by a number unless it comes at the beginning of a sentence, e.g.:
    "The results are depicted in Fig. 5. Figure 9 reveals that...".’ We fixed the occurrences
    of Figure that should have been Fig.**

Fig. 3 and other occurrences in Figures - units should not be in italics.

    **Fixed the italics in all figures.**

L112 - the first “($Q_p$)” is unnecessary.

    **Removed $Q_p$.**

L112 - Consider using a different letter for coefficients, or being consistent with the subscripts, or
otherwise reducing the potential for confusion; currently there are two $C_x$’s for concentrations
(L95), a $C_p$ for the peak discharge coefficient (L112), and a C for the runoff coefficient (L115).

    **We now use S for the stream concentration and D for the injection dye concentration,
    and $C_p$ for peak discharge coefficient, and $C_R$ for runoff coefficient.**

L137 - delete “and” and add a comma before phi

    **Changed as suggested.**

L152 - Figure 4 and its description could be easier to follow. In the text, figure, and caption can the same terms be used for "turbulent-underwater" and "open-channel" melt be used? The description of moulin deformation modelled as viscous and elastic needs to be separated from the shear deformation of the ice modelled using Glen's flow law. Can you mention that the former is modelled using a Maxwell model. Overall the description of the model needs to be expanded to briefly introduce all of the components without the reader needing to refer to Andrews et al. (2022) to make sense of the modelling approach.

**We will improve the Moush model description using these suggestions.**

Figure 5 caption - In "(d-e) Moulin shape evolves with surface input." does this mean that other processes affecting moulin shape were excluded?

**We changed to "Moulin shape is free to evolve through time", as opposed to simulation Fa-Fc which have a fixed moulin radius through the simulation.**

Should "Simulations EMa"be singular? And should "head of oscillation" be "oscillation of head"?

**Changed to "Simulation EMa:..."**

Generally interpretation of figures should be left to the main text and not included in the caption.

**Removed "This simulation illustrates that a large volume is needed to reproduce realistic head amplitude." from the caption in Figure 5.**

Figure 5 c,e,g y-axis labels - Symbol z has not been defined in the text, and in Figure 4 "Height above the bed (m)" is written out in full.

**Updated figure 4 to show z and x instead, and added "The horizontal position (x) varies along the moulin height above the bed (z)." in the caption of Figure 4.**

L193 - specify the sim for the fixed 5 m model run.

**We added "(Sim Fb)" in : "For our model to reproduce measured head oscillations (black lines), a fixed moulin radius of 5 m (Sim Fb) was required …"**

L206 - plural 'radii"

**Corrected radius to radii.**

L214 - Specify the sim before referred to.

**We added "with Sim EMb" in : "While the general diurnal range of moulin head is reproduced with Sim EMb, the match between the simulated and measured head is imperfect."**

L254 - Specify Sims E1a-d.

**Moved "(Sim EMa)" after "without baseflow" to avoid the confusion.**

L259 - "of oscillation" could be omitted.

**Removed "of oscillation".**

Fig. 8 - Units incorrectly in italics.

**Removed italic in all units in all the figures.**

L290 - is it necessary to specify normalized here when discussing in general terms? Is this $A^*_{in}$?

**Yes, the "normalized" is important to specify, because the ratio between the mean meltwater input and the diurnal range controls the variability, not the diurnal variability itself. And yes, it is $A^*_{in}$. We added the symbol.**

Table 3 - Can you use a narrower dash to indicate that a variable is unitless to avoid ambiguity with the minus sign, or perhaps just leave the units for unitless variables blank as on L353? (A narrower unitless symbol is used on Figure 9).

**Made the dash narrower as suggested**

L333 - add "moulin" after "FOXX"

**Added moulin as suggested**

L380 - add "respectively" after "base flow"

**Modified as suggested**

L477 - revise to avoid the apparent contradiction in dismissing "subsurface inputs" which would dismiss "basal inputs".

**Added "shallow" in : "With surface and shallow subsurface external inputs dismissed at our site…"**

L468 - the following sentence needs revising to make sense "with only a small portion of the water can"

**Changed the verb tense of the verb "to travel": "In reality, before it can make it to the subglacial channel system, some portion of this available basal melt is stored in the inefficient portion of the drainage system with only a small portion of the water traveling through the linked cavity system …"**

**Additional References**

Gordon S et al. (2001) Borehole drainage and its implications for the investigation of glacier hydrology: experiences from Haut Glacier d'Arolla, Switzerland. Hydrological Processes 15, 797-813. doi:10.1002/hyp.184.

Gulley J (2009) Structural control of englacial conduits in the temperate Matanuska Glacier, Alaska, USA. Journal of Glaciology 55, 681-690.

Holmlund P and Hooke R (1983) High-water pressure events in moulins, Storglaciaren, Sweden. Geografiska Annaler 65A, 19-25.

Hubbard B et al. (2021) Borehole-Based Characterization of Deep Mixed-Mode Crevasses at a Greenlandic Outlet Glacier. AGU Advances 2, e2020AV000291. doi: 10.1029/2020AV000291.

Luthi MP et al. (2015) Heat sources within the Greenland Ice Sheet: dissipation, temperate paleo-firn and cryo-hydrologic warming. The Cryosphere 9, 245-253. doi:10.5194/tc-9-245-2015.

Meierbachtol T, Harper J and Humphrey N (2013) Basal drainage system response to increasing surface melt on the Greenland Ice Sheet. Science 341, 777–779. doi:10.1126/science.1235905.

Vieli A et al. (2004) Short-term velocity variations on Hansbreen, a tidewater glacier in Spitsbergen. Journal of Glaciology 50, 389-398. doi: 10.3189/172756504781829963.

**References**

Andrews, L. C., Poinar, K., & Trunz, C. (2022). Controls on Greenland moulin geometry and evolution from the Moulin Shape model. *The Cryosphere, 16*(6), 2421–2448. https://doi.org/10.5194/tc-16-2421-2022

Chandler, D., Wadham, J. L., Nienow, P. W., Doyle, S. H., Tedstone, A. J., Telling, J., Hawkings, J., Alcock, J. D., Linhoff, B., & Hubbard, A. (2021). Rapid development and persistence of efficient subglacial drainage under 900 m-thick ice in Greenland. *Earth and Planetary Science Letters, 566*, 116982. https://doi.org/10.1016/j.epsl.2021.116982

Covington, M. D., Gulley, J. D., Trunz, C., Mejia, J., & Gadd, W. (2020). Moulin Volumes Regulate Subglacial Water Pressure on the Greenland Ice Sheet. *Geophysical Research Letters, 47*(20). https://doi.org/10.1029/2020GL088901

Gordon, S., Sharp, M., Hubbard, B., Willis, I., Smart, C., Copland, L., Harbor, J., & Ketterling, B. (2001). Borehole drainage and its implications for the investigation

of glacier hydrology: Experiences from Haut Glacier d'Arolla, Switzerland. *Hydrological Processes*, *15*(5), 797–813. https://doi.org/10.1002/hyp.184

Gulley, J., Benn, D. I., Müller, D., & Luckman, A. (2009). A cut-and-closure origin for englacial conduits in uncrevassed regions of polythermal glaciers. *Journal of Glaciology*, *55*(189), 66–80. https://doi.org/10.3189/002214309788608930

Holmlund, P., & Hooke, R. LeB. (1983). High Water-Pressure Events in Moulins, Storglaciären, Sweden. *Geografiska Annaler. Series A, Physical Geography*, *65*(1/2), 19–25. https://doi.org/10.2307/520717

Hubbard, B., Christoffersen, P., Doyle, S. H., Chudley, T. R., Schoonman, C. M., Law, R., & Bougamont, M. (2021). Borehole-Based Characterization of Deep Mixed-Mode Crevasses at a Greenlandic Outlet Glacier. *AGU Advances*, *2*(2), e2020AV000291. https://doi.org/10.1029/2020AV000291

Lüthi, M. P., Ryser, C., Andrews, L. C., Catania, G. A., Funk, M., Hawley, R. L., Hoffman, M., & Neumann, T. A. (2015). Heat sources within the Greenland Ice Sheet: Dissipation, temperate paleo-firn and cryo-hydrologic warming. *The Cryosphere*, *9*(1), 245–253. https://doi.org/10.5194/tc-9-245-2015

Mejia, J. Z., Gulley, J. D., Trunz, C., Covington, M. D., Bartholomaus, T. C., Xie, S., & Dixon, T. H. (2021). Isolated Cavities Dominate Greenland Ice Sheet Dynamic Response to Lake Drainage. *Geophysical Research Letters*, *48*(19), e2021GL094762. https://doi.org/10.1029/2021GL094762

Poinar, K. (2016). *The influence of meltwater on the thermal structure and flow of the Greenland Ice Sheet* [University of Washington]. https://digital.lib.washington.edu/researchworks/handle/1773/35062

---

## Author Comment (AC2)

**Author response to comment on "Observed and modeled moulin heads in the Pâkitsoq region of Greenland suggest subglacial channel network effects"**

Response to Anonymous Referee #2

**The authors' replies are in bold blue**

**General comments**

**We thank Referee #2 for their comprehensive review and for their useful comments.**

This is an interesting and well-written paper that tackles the problem of how to interpret moulin water pressure (head) records. The paper addresses an important topic because moulins are a less inconvenient way of observing subglacial hydrology, than methods such as boreholes.

The main focus of the paper is a suite of experiments with the MouSh model coupled to a subglacial channel model. The experiments attempt to match a moulin head record obtained in the well-studied Paakitsoq region of Greenland. The crux of the problem seems to be creating enough damping in the simulation, so that simulated diurnal head variations remain small enough. A match can be optimised by specifying a suitable subglacial base flow or a very large moulin shaft. Although the experiments themselves are interesting conceptually, there are two very significant weaknesses that limit how useful this study is in its present form.

First, to match the simulated moulin head with observations, the authors need to add subglacial water (base flow). Although this is of course quite reasonable when modelling a moulin that joins a wider scale drainage system, it's not clear how much of the discrepancy between modelled/observed moulin head is due to (i) the subglacial base flow and subglacial channel model, (ii) uncertainties in the moulin model, and (iii) the rather poorly prescribed melt input. Taking these in turn:

In the case of base flow, it is confusing that it is prescribed as a flow rate (m3/s) even though that is on first impression an obvious choice. The moulin head is a proxy for subglacial water pressure (Pw), not flow, and even though these quantities are related, prescribing flow rather than Pw presumably requires some speculation of drainage system characteristics to calculate Pw in an extra step. In fact without any base flow or stream input at all, the moulin head could simply reflect changes in Pw driven entirely non-locally, provided the moulin remains hydrologically connected. This aspect needs to be clearer so we know what assumptions are needed and how are the associated parameters constrained. It is also not obvious how Qout is

calculated at the bottom of the moulin, nor how the subglacial channel model is driven in the context of wider-scale drainage evolution through the season.

> **The moulin head is indeed a proxy for the water pressure in the subglacial channel, however that water pressure (or head) is related to water flow: the moulin head is a function of the water input in the moulin (m3/s), the water output in the subglacial channel (m3/s), and the cross-sectional area of the moulin (m2). The subglacial baseflow is added to the subglacial output and does not transit through the moulin. This baseflow influences the moulin head and the subglacial cross-sectional area by modifying Qout only. We will add this description in the Methods section to make it clear how we add baseflow to the model.**

A minor related point: the upper limit for baseflow of 5 m3/s in the experiments seems very low for a typical Greenland catchment or even a small part of one.

> **This may be, but our moulin and its surface catchment are quite small: the catchment is just 0.24km$^2$. For our specific simulations, 5 m$^3$/s is the equivalent of 20 additional moulins being fed the same surface melt upstream of our moulin, and directly connected to a subglacial channel which would have to flow directly into the subglacial channel formed by our moulin. Moulin density is often lower than 1/km$^2$ in Greenland (Banwell et al., 2016). In addition, the baseflow parameter is a raw parameter and it is not yet clear how this parameter is constrained. A very large water flow in the main subglacial channel might only transfer a smaller amount of water to a nearby moulin through its tributary channel. Our simulation results worked best with a 2m$^3$/s baseflow. We will add a discussion point in section 5.4 .**

For the moulin model we need to remember that MouSh is effectively unvalidated, even though it is quite a detailed model. Inevitably there are many unknowns here – for example, my understanding is that MouSh assumes an initial moulin that is a vertical cylinder, which I suspect is far from reality given that most moulins appear to form as hydrofractures in Greenland, which are initially planar. Anybody that has tried lowering sensors into a moulin can guess they are not vertical shafts! Limited exploration confirms that. Perhaps several moulins are connected englacially within an initially planar fracture zone, before even reaching a subglacial channel. Almost like a 'distributed' englacial drainage system that might also produce the dampened head record sought by the authors. Of course this is speculative, maybe moulins are vertical shafts after all, but in the experiment we should account for our lack of knowledge in this respect.

> **We recognize that the initial shape of the moulin in the MouSh is simplified as a cylinder and that more complex shapes are more likely to originate from hydrofracture, and that the upper part of the moulin does have plunge pools created by the surface stream which creates a challenge for lowering instruments. The possibility of "planar storage" is an interesting thought that is a possibility for Greenland moulins in general. Indeed, observations of other moulins near our field site showed some lines of moulins that could be connected with each other through a larger crevasse, and some did have visible crevasses associated with them. However, neither applied to the moulin we study here. We also observed small early-season moulins starting in small crevasses. As mentioned in our answer to Reviewer 1, "To substantially influence the**

**water level fluctuation, the cross-sectional area of the englacial void would have to be about 80m², so for example a crevasse of 10m long and 8 cm wide, at a depth of 150-200m."  However, at our site, surface observation shows no crevasses. In addition, Andrews et al. (2022) demonstrate that in most cases, initialization processes smaller than two weeks are overwritten by creep processes. While it seems unlikely, we cannot discard the potential for planar storage at our site, therefore, we will expand the discussion section to discuss the potential for englacial storage from moulins connected through fractures in section 5.2.2.**

Melt input: this is the last important source of uncertainty, and it seems very much brushed under the carpet in Section 3.1.1. Even the very short (2-day) observed time series is not well represented by either the 'Modeled' or 'Idealized' time series (Fig 3), and the extended parts of the Q time series shown in Fig 3 do not follow the trend of the steam water level, which is discouraging. The vast majority of the melt input time series is extrapolated outside of this short (poorly matched) tuning window. I strongly disagree with their "good confidence in our derived runoff values R and thus our model forcing Qin,model".

**We recognize that the direct stream measurements used to calibrate the routing model have a short duration and the modeled meltwater input might appear not perfectly constrained.**

**Though the length of the measured time series we use to calibrate our stream discharge is only two days, it still reduces the uncertainty from what we would obtain with the met station measurements alone. Calculated meltwater inputs can be orders of magnitude over or under-estimated with melt models (McGrath et al 2011, Smith et al. 2017). Melt models calculate an expected melt rate (m. w.e.). To convert this to runoff, the drainage basin, the ice density, and the fraction of water refreezing are minimally constrained parameters.**

**Because of the extremely limited supraglacial stream discharge and moulin head measurements on the Greenland Ice Sheet, this is the only study that can use direct stream measurements simultaneously with direct measurements in moulins. Please note that a key finding in this manuscript is that our relative diurnal range of meltwater input falls in the range of other measured meltwater stream fluxes in Greenland (the usual diurnal range ranges between 1-2x the mean discharge - Table 3). Figure 8a shows that in the absence of baseflow, the diurnal range would need to be less than a quarter of the mean discharge, which has not been observed in Greenland (Table 3). Therefore, uncertainties in the meltwater model calibration are negligible compared to those values.**

**We realize that our description of the meltwater input may have been misleading, and we will remove our statement of good confidence and expand 3.1 to include more details about the meltwater input model calibration, its limitation, and potential errors.**

**Regarding the stream water level: Stream water level is controlled by the cross-sectional area of the supraglacial stream, as well as the distance between the instrument and the bed of the stream, which evolves through time. The lowering trend of the stream water level time series after day 206 is artificial; it is due to the deepening of the bottom of the stream beneath our sensor after we left the field. In**

**addition, the curvature of the daily amplitude of the stream water level cannot be directly compared with the stream discharge. Usually, hydrology methods use a simple regression curve to interpolate corresponding discharge for specific water levels. Unfortunately, we cannot use those methods because of the constant lowering of the stream bed and the cross-sectional area variation over time. However, the stream water level does reflect the increase in supraglacial stream discharge caused by the cloud coverage. We will describe the stream gauge better in the Methods section.**

I also wonder why most simulations (Tab 2) use the modelled Qin, even though it looks like a worse fit than the idealized?

**The idealized input might be a good fit for the specific days we measured discharge, however it does not represent the seasonal variability of the meltwater input throughout the season as well as the modeled $Q_{in}$ does. This is why we use the modeled $Q_{in}$ for the first sets of simulations.**

This poor match is not necessarily a problem when constraining a model with observations, if it is clearly acknowledged and if the uncertainty it introduces is more rigorously accounted for.

**We will improve the Methods section by further describing the meltwater input calibration and providing a range of uncertainty based on the 2017 melt season.**

I think to address this first weakness we would need some detailed error analysis before any comparison between the simulations and observed head time series can be interpreted in a useful manner. I would envisage an ensemble (e.g. Monte Carlo or latin hypercube sampling) or more advanced statistical approach to account for the very many uncertainties in the MouSh model, the treatment of baseflow, and the stream inputs. The long discussion section (which is currently very speculative without even a basic error analysis) should then focus on how much of the record can be confidently interpreted in terms of real variations in subglacial water pressure (or drainage characteristics), and how much cannot be untangled from uncertainty in the simulation and inputs.

**This is an interesting comment. However, for this study specifically, variations in meltwater input do not affect the main results of this manuscript. As mentioned above, even if there are some uncertainties with the tuning of the melt model and with the melt model itself, it reproduces relative meltwater amplitude in a similar range as other measured streams in Greenland. The simulations in Figure 8a demonstrate that the amplitude of oscillation relative to the mean discharge needs to be ten times smaller than the measured diurnal variability in supraglacial streams in Greenland (Table 3). We will clarify this point in the manuscript.**

**Regarding the treatment of baseflow, we think an involved ensemble approach would be incongruent with the simplistic nature of typical baseflow. Our forward-model approach, where we prescribe constant, sinusoidal, and lagged-sinusoidal baseflow (i.e. a maximum of 3 parameters: mean, range, and phase) is better suited here.**

**Finally, the reviewer has a fair point that the MouSh model contains many processes that have inherent uncertainties. Rather than rehash the extensive sensitivity testing done by Andrews et al. (2022) -- 24 pages! -- we instead suggest that we will discuss the effects of the more poorly constrained aspects of MouSh (e.g. above-water-line melt, viscous deformation, below-water-line melt) in the Discussion. Note that we already address potential biases from the above-water-line melt module in Section 5.2.1.**

The second weakness is its relevance, which of course I acknowledge is limited by where field data are available. In Fig 5 it is evident that the study is conducted mid melt season when there is relatively little variation in moulin head (range looks like 250 to 400m, but much of period close to 300m), and as such Pw is always well below the ice overburden pressure. In fact the authors choose a period mid-season when h is around 60% of H and varies diurnally by about 10%. In these conditions we would expect the moulin head to have a minimal effect on ice motion. Probably there are some data for this region that could answer that more precisely. What the community needs more, I would suggest, is for the study to simulate the early season formation of moulins as part of efforts to simulate the duration/extent of ice acceleration in spring. What role do moulins play in the evolution to channelised drainage & associated ice deceleration? It's not clear how the results would help that aim, at the moment. However, some simulations of moulin head very early in the season could provide valuable pointers for interpreting moulin pressure records in the more dynamically important part of the cycle, or in thicker ice. Similarly, experiments simulating extreme melt pulses mid or late season could be useful. But, related to the first point, we would need to know how the subglacial channel model is driven by / coupled with the larger-scale hydrological evolution.

**We agree that the early melt season would be very interesting to investigate. Because we do not have field data from the beginning of the melt season and because the model limitation shows that we are missing a basal influx component, investigating the early season might be off-topic for this particular research. We thank the reviewer for the thoughtful ideas for future research, which we agree with and would be excited to enact in the future. We believe that our results from the melt season are relevant, as our work combines, for the first time, field data constraints for meltwater inputs and moulin water level, as well as moulin shape constrained by a physically based model, and shows that for this particular case, englacial storage might not be the answer to moulin water level damping.**

Overall I think this could go one of two ways – (i) keep the focus on the link with available field observations, by carrying out some detailed error analysis, or (ii) accept that the field observed melt inputs are perhaps too limited/uncertain at present, and instead focus on a conceptual study that is not tied to that one location and can explore controls on head variations across a wider (more interesting) range of sites/conditions. Either of these directions could turn this into a really useful study to help interpret or design experiments with moulin water pressure records.

**We thank the reviewer for the recommendations. Suggestion (ii) is included in Andrews et al. 2022. This present manuscript instead, focuses on a region where we have the most in-situ constraints, in order to identify some system properties which we may be able to broaden more regionally. For suggestion (i) we think that detailed**

**error analysis is not necessary. This is because the field observations of stream flow, while uncertain in an aleatoric sense, is a far lesser contributor than the epistemic uncertainty inherent to our simple one-moulin, one-channel model. The calibrated meltwater input greatly reduces the uncertainty and this uncertainty is orders of magnitude lower than the uncertainty in the subglacial processes. The moulin size error is also an order of magnitude lower than what would be required to dampen the moulin head; Andrews et al. 2022 conducted a detailed sensitivity analysis of the MouSh model that shows this. We recognize that some of the meltwater input descriptions were badly formulated and created some confusion. We will improve our description of the meltwater input model and demonstrate how the uncertainty does not matter so much because the general ratio diurnal range/mean discharge is constrained enough for the purpose of this manuscript.**

**Minor comments from the introduction**

L2: I believe water pressure also influences motion in some marine terminating glaciers (not just land terminating).

**Subglacial flow does also affect marine terminating glaciers. However, in this manuscript, our focus is on land terminating glaciers for 2 main reasons: 1. Subglacial water pressures are the main driver of ice motion and 2. the moulin we study is in a land terminated area. In marine terminating glaciers, the relationship between tidal influence, subglacial water pressure, and freshwater plumes represents a complex environment that we believe cannot be compared with our current study. So, we have left the mention of marine-terminating glaciers out of our abstract.**

L19 I think some observation based papers showed the influence of temporal melt inputs, before Schoof 2010 – worth to cite these here too.

**We change the sentence in the Introduction "... and temporal (Bartholomaus et al., 2008; Iken & Bindschadler, 1986; Schoof, 2010) supraglacial meltwater input variability can …"**

L28 again there are earlier papers describing drainage evolution in Greenland (Bartholemew et al 2011 EPSL?), and of course even earlier elsewhere (1980s/90s work on alpine glaciers).

**We add "While earlier studies suggested that the efficiency of the subglacial drainage system controls the seasonal pattern of velocities (Bartholomew et al., 2010; Iken & Bindschadler, 1986), field observations of basal water pressures in Greenland demonstrated instead the prominent role of the weakly connected drainage system (Andrews et al., 2014; Hoffman et al., 2016)."**

In general there is a lot of self citation in the intro that could be expanded to include earlier work from other groups.

**We did our best to cite the relevant literature, but doubtless there are papers we do not know about.  We will spend some time searching for earlier work to cite.**

Fig 8 confusing that red dots can be either observations or simulations. Can sims not be blue or some other colour?

**Changed the sim to orange instead of red and updated the figure caption to match the change.**

References:

[revised manuscript text omitted]

---

## Editor Decision (ED1)

Editors report for Trunz et al. MouSh paper.

Dear authors,

Thank you for your thorough response to the revisions. Both reviewers agree that the paper is of good scientific merit, although one is more critical of the overall study design. On reading your response and revision, I think there is opportunity to satisfy both reviewers and ensure your results are publishable by slightly changing the narrative of the paper.

The paper is presented as a generalised study of moulin processes, using (highly valuable and hard won!) field data to validate your modelling experiment. However, the moulin monitored in the field is actually pretty small in the context of Greenland moulins. I think that the study would be more applicable, and acceptable to Reviewer 2 if you can make it clearer in the text that this is likely a small moulin that feeds a lower order subglacial channel, that may in turn feed into the wider network. I think this would address some of the difficulties that Reviewer 2 has with the study – it is representative of a certain aspect of the subglacial drainage system, but is not what we would generally think of when visualising the surface-subsurface connectivity for the main trunk of the drainage system. Your conceptual model figure is really valuable to illustrate this, and you do acknowledge it within the text already, but I think putting this understanding up front would be beneficial. The study is not aiming to represent moulin processes generally, but to understand moulin evolution in a small, upper catchment that feeds a larger subglacial network. This would then tally with the high subglacial base flow necessary to match your model results with the field data.

I disagree with Reviewer 2 that the study is of limited relevance: anything that helps us to understand this very complex system is valuable, but we must be cautious in overstating the scope – just because there isn't much data available, it doesn't mean that any data we have can be used to understand the whole system! Please see if you can adjust the narrative somewhat to satisfy these concerns (without requiring the full error analysis suggested by Reviewer 2), and also address the minor amendments below.

Abstract: please add 'small' as a preface to the moulin, or something else that flags that this is not a Greenland giant draining a large area, but part of a wider network of surface-subsurface connections.

Introduction: in lines 66-70, can you make it clear that this is a moulin-dense catchment and you are simulating just one small moulin that is likely part of a wider network

Methods: please could you add more clarity on the Qin modelling – your model output doesn't seem to capture the measured. You discuss some of the uncertainties associated with this, but I think a little more might be useful to address some of Reviewer 2's concerns. You add a little in L160-165, but I think more discussion of why the Qin does not capture the full range of the measured Q would be beneficial.

L375 + L377: reference bracketing incorrect

L384: reference punctuation incorrect

L404-409: this seems like a really important point which helps to address the concerns of reviewer 2: your moulin is small, formed that year, and is relatively small. This means the behaviour you simulate is realistic, but only for this scenario of hydrology.

L446: slightly unclear meaning

L558-9: this seems like a very large area feeding a very small stream and small moulin, especially given you note that there was a relatively high moulin density in the area. Please could you clarify whether this catchment feeds your single moulin, or whether it feeds the multiple streams and moulins in the region?

Conclusions:

L570: 'Our results suggest that the moulins on the Greenland Ice Sheet require larger inputs than surface meltwater alone to keep their subglacial channels large enough to accommodate the observed wide diurnal range of surface input.'

The first line of the conclusion reads as if this moulin is representative of all moulins in Greenland, which you acknowledge in the paper and in your response that it is not. I would like the first sentence of the conclusions rephrased to better inform the reader of the applicability of your experiment.

Thank you for your submission to The Cryosphere and your patience in receiving these comments: I've had my head down a moulin in Greenland.

Dr Liz Bagshaw, Editor

---

## Author Response (AR2)

**Author response to editor's report on "Observed and modeled moulin heads in the Pâkitsoq region of Greenland suggest subglacial channel network effects**

**We thank the editor for their time and comments on the paper and review. The authors' replies are in bold blue**

Editors report for Trunz et al. MouSh paper.

Dear authors,

Thank you for your thorough response to the revisions. Both reviewers agree that the paper is of good scientific merit, although one is more critical of the overall study design. On reading your response and revision, I think there is opportunity to satisfy both reviewers and ensure your results are publishable by slightly changing the narrative of the paper.

The paper is presented as a generalised study of moulin processes, using (highly valuable and hard won!) field data to validate your modelling experiment. However, the moulin monitored in the field is actually pretty small in the context of Greenland moulins. I think that the study would be more applicable, and acceptable to Reviewer 2 if you can make it clearer in the text that this is likely a small moulin that feeds a lower order subglacial channel, that may in turn feed into the wider network. I think this would address some of the difficulties that Reviewer 2 has with the study – it is representative of a certain aspect of the subglacial drainage system, but is not what we would generally think of when visualising the surface-subsurface connectivity for the main trunk of the drainage system. Your conceptual model figure is really valuable to illustrate this, and you do acknowledge it within the text already, but I think putting this understanding up front would be beneficial. The study is not aiming to represent moulin processes generally, but to understand moulin evolution in a small, upper catchment that feeds a larger subglacial network. This would then tally with the high subglacial base flow necessary to match your model results with the field data.

I disagree with Reviewer 2 that the study is of limited relevance: anything that helps us to understand this very complex system is valuable, but we must be cautious in overstating the scope – just because there isn't much data available, it doesn't mean that any data we have can be used to understand the whole system! Please see if you can adjust the narrative somewhat to satisfy these concerns (without requiring the full error analysis suggested by Reviewer 2), and also address the minor amendments below.

- **We have added "... and is coherent with our small moulin likely connected to a low order subglacial channel." at the end of the abstract.**
- **Made the sentence more clear L66-67 that it is a small connected moulin "In this study, we investigate the hydrodynamics in the englacial-subglacial system of a small single moulin in a moulin-dense catchment using a single-conduit subglacial model coupled with the Moulin Shape (MouSh) englacial hydrology model."**
- **We modified the first sentence of the discussion to: "In this study, we provide the first comparison of modeled hydraulic head in a shape-evolving moulin to direct field measurements of a small subglacially connected moulin in Greenland.**
- **We modified the conclusion as suggested below**

Abstract: please add 'small' as a preface to the moulin, or something else that flags that this is not a Greenland giant draining a large area, but part of a wider network of surface-subsurface connections.

**Added "small" in front of moulin**

Introduction: in lines 66-70, can you make it clear that this is a moulin-dense catchment and you are simulating just one small moulin that is likely part of a wider network

**Modified the sentence to:**

**"In this study, we investigate the hydrodynamics in the englacial-subglacial system of a single moulin in a moulin-dense catchment using a single-conduit subglacial model coupled with the Moulin Shape (MouSh) englacial hydrology model." L65-68**

Methods: please could you add more clarity on the Qin modelling – your model output doesn't seem to capture the measured. You discuss some of the uncertainties associated with this, but I think a little more might be useful to address some of Reviewer 2's concerns. You add a little in L160-165, but I think more discussion of why the Qin does not capture the full range of the measured Q would be beneficial.

**We reorganized the Qin modeling section so that description of the model and limits are more separate. It seems that in the previous organization it was not clear where we were discussing limitation. We also clarified that the moulin - suglacial channel model is not sensitive to small variations of modeled meltwater input on a daily basis or small variations of peak. The Moush model output is sensitive to the multiday ratio between amplitude and mean discharge.**

L375 + L377: reference bracketing incorrect

**Fixed the parenthesis for both citations.**

L384: reference punctuation incorrect

**Removed the incorrect punctuation.**

L404-409: this seems like a really important point which helps to address the concerns of reviewer 2: your moulin is small, formed that year, and is relatively small. This means the behaviour you simulate is realistic, but only for this scenario of hydrology.

L446: slightly unclear meaning

**Changed the sentence to: "However, at our site, there is no surface crevasse passing through or close to the moulin …"**

L558-9: this seems like a very large area feeding a very small stream and small moulin, especially given you note that there was a relatively high moulin density in the area. Please could you clarify whether this catchment feeds your single moulin, or whether it feeds the multiple streams and moulins in the region?

**Thanks for the comment. Since we are looking at the subglacial catchment potentially draining water connected to the moulin we are investigating, we included the area all the way up to the divide. This is why this is a large area. We've added text to emphasize that this is a SUBglacial area, not a SUPRAglacial area (where, indeed, 2000 km2 would be very large).**

Conclusions:

L570: 'Our results suggest that the moulins on the Greenland Ice Sheet require larger inputs than surface meltwater alone to keep their subglacial channels large enough to accommodate the observed wide diurnal range of surface input.'

The first line of the conclusion reads as if this moulin is representative of all moulins in Greenland, which you acknowledge in the paper and in your response that it is not. I would like the first sentence of the conclusions rephrased to better inform the reader of the applicability of your experiment.

**We modified the first sentence to be more specific towards our moulin, rather than generalizing to all moulins in Greenland: "Our results suggest that the moulin we instrumented requires larger inputs than surface meltwater alone to keep its subglacial channels large enough to accommodate the observed wide diurnal range of surface input."**

Thank you for your submission to The Cryosphere and your patience in receiving these comments: I've had my head down a moulin in Greenland.

Dr Liz Bagshaw, Editor

---

## Author Response (AR3)

**Author response to editor's comments on "Observed and modeled moulin heads in the Pâkitsoq region of Greenland suggest subglacial channel network effects"**

**We thank the editor for their comments and support during the review process of this paper.**

**The authors' replies are in bold blue**

Editor's comment:

Thank you for your thorough responses to suggested changes to the paper. I am satisfied that the reframed argument and justification of the assumptions made is sufficiently clear to warrant publication. I make two final suggestions:

1. Move Figure 10 higher up the discussion to aid interpretation of the text.

**We moved the figure higher up in the discussion.**

2. Basal melt rates: there is a more recent article published by Karlsson et al. 2021 (https://www.nature.com/articles/s41467-021-23739-z) that discusses basal melt rates and potential melt fluxes. It could be a worthwhile addition to the basal melt discussion section.

**We added Karlsson et al. 2021 recent results on basal melt to the discussion (L535-538).**

**We also had a couple small lingering co-author edits in the abstract that I incorporated for that last version.**